# Multiplexed representation of others in the hippocampal CA1 subfield of female mice

Xiang Zhang [1,2,6], Qichen Cao[1,2,3,6], Kai Gao[1,2,3], Cong Chen[1,2], Sihui Cheng[1,2,3], Ang Li [1,2,3], Yuqian Zhou [1,2], Ruojin Liu[1,2,3], Jun Hao[1,2,3], Emilio Kropff[4,7] ✉ & Chenglin Miao [1,2,3,5,7] ✉

Hippocampal place cells represent the position of a rodent within an environment. In addition, recent experiments show that the CA1 subfield of a passive observer also represents the position of a conspecific performing a spatial task. However, whether this representation is allocentric, egocentric or mixed is less clear. In this study we investigated the representation of others during free behavior and in a task where female mice learned to follow a conspecific for a reward. We found that most cells represent the position of others relative to self-position (social-vector cells) rather than to the environment, with a prevalence of purely egocentric coding modulated by context and mouse identity. Learning of a pursuit task improved the tuning of social-vector cells, but their number remained invariant. Collectively, our results suggest that the hippocampus flexibly codes the position of others in multiple coordinate systems, albeit favoring the self as a reference point.

The rodent cognitive map provides an internal representation of self-position within an environment through the activity of neurons such as hippocampal place cells and entorhinal grid cells[1–5]. In addition, evidence suggests that the same networks encode the location of external bodies, such as objects and conspecifics[6–11]. Object-vector cells in the hippocampus encode the position of an object relative to the animal, irrespective of the position of the object within the environment[12]. One synapse upstream, the medial entorhinal cortex has object-vector cells with similar characteristics[13], while the lateral entorhinal cortex has been shown to represent the relative orientation of objects using a purely egocentric system of coordinates that rotates with the animal's head[14]. Egocentric representations can code for goal locations in the hippocampus[15–17], and have been reported in other brain areas[18–21].

Precise representation of others during social interactions poses additional challenges since conspecifics, unlike objects, can rapidly and unpredictably change their location. Recent groundbreaking work has shown that the hippocampus of a passive observer codes the position of a conspecific performing a stereotypical task[10,11]. However, since in this kind of paradigm the observer is mostly still, it is less clear

whether the location of the other is being represented relative to the observer, to the environment, or through a combination of both perspectives. More generally, whether and how these networks represent the location of others during free and spontaneous social interactions, or during social tasks in which both animals play simultaneously active roles, remains largely unknown. Addressing these questions requires the systematic examination of cell activity in multiple simultaneous reference frames, including egocentric and allocentric perspectives.

In this work, we show single-unit calcium activity recordings taken from the CA1 subfield of mice foraging in open field environments in the presence of conspecifics. Our analyses focus on four reference frames. First, the classical allocentric representation of self-position in which place cells were characterized. Second, the allocentric representation of others introduced to study social spatial coding[10,11]. In addition, we hypothesize the representation of the location of others relative to self-position in either a fixed coordinate system or in one that rotates together with the animal's head, two possibilities that previous experiments were not designed to test for. We also present recordings in a

[1]State key laboratory of Membrane biology, School of Life Sciences, Peking university, Beijing 100871, China. [2]PKU-IDG/McGovern Institute for Brain Research, Peking university, Beijing 100871, China. [3]Peking-Tsinghua Center for Life Sciences, Beijing 100871, China. [4]Leloir Institute/IIBBA-CONICET, Buenos Aires, Argentina. [5]Chinese Institute for Brain Research (CIBR), Beijing, China. [6]These authors contributed equally: Xiang Zhang, Qichen Cao. [7]These authors jointly supervised this work: Emilio Kropff, Chenglin Miao. ✉e-mail: ekropff@leloir.org.ar; chenglin.miao@pku.edu.cn

novel behavioral test in which a mouse learns to chase a conspecific for a reward. Using this task we assess how the representation of others relates to performance in a social task with simultaneously active roles.

## Results

### CA1 cells represent the position of conspecifics relative to self-position

To study social spatial representations in the hippocampus, we recorded calcium events (binarized with a cutoff of 3 standard deviations) from GCaMP6s-expressing unique CA1 neurons, using miniaturized endoscopes, while mice foraged in a 70 cm-diameter circular open field for 10 min together with a familiar conspecific (a cage mate; Fig. 1 a–c, Supplementary Fig. 1 and Supplementary Movies 1–2; one session per pair of mice).

For each recorded cell we constructed four spatial maps, using [x,y] coordinates of either the self, the other or the other relative to the self (Fig. 1d–e and Supplementary Fig. 2). Maps for the relative position used allocentric or egocentric coordinates, in the latter case rotating with the head of the imaged mouse. With these definitions, the space representing all possible relative positions (the 'effective arena') exhibited dimensions twice as large as those of the physical arena (4 times in terms of area), so spatial bins for relative position were also twice as large (Supplementary Fig. 3a). By comparing the spatial information content in these maps with the distribution resulting from 1000 temporal shuffles of the calcium events, we classified each cell as self-place cell (selfPC), social-place cell (socialPC), allocentric social-vector cell (alloSVC) or egocentric social-vector cell (egoSVC), allowing for cells to be included in multiple categories simultaneously (Fig. 1e

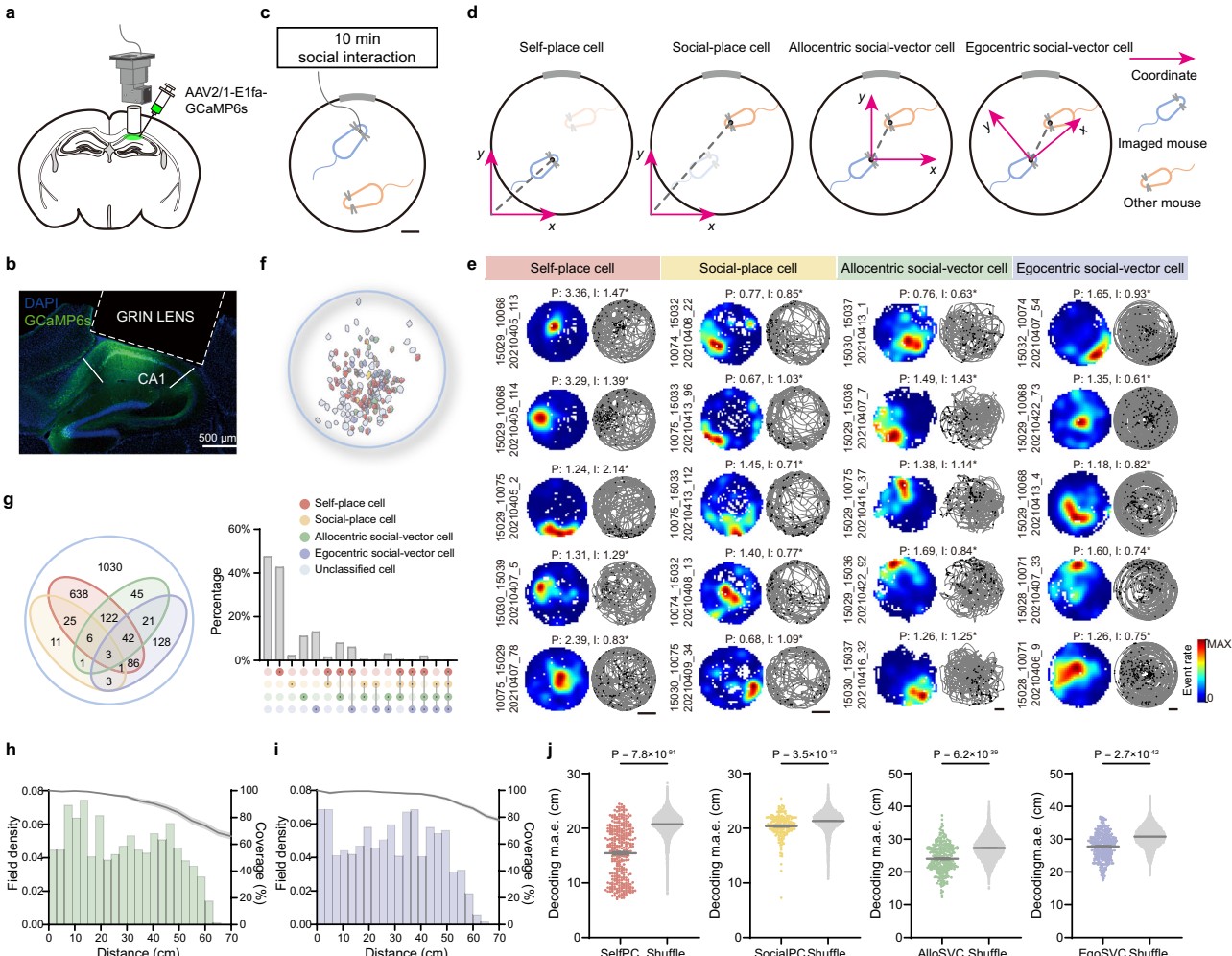

**Fig. 1 | Multiplexed social coding in CA1 favors social-vector representations.**
**a** Schematic of AAV2/1-EF1a-GCaMP6s virus injection, GRIN lens implantation and miniscope imaging. **b** A representative example of histological slice showing GCaMP6s expression (green), DAPI staining of nuclei (blue) and the trace of a GRIN lens (dashed lines). Scale bar: 500 μm. **c** Diagram of the experimental paradigm. An imaged mouse (blue) and a social partner (orange) interacted freely in an open field arena (18 imaged mice, 2162 imaged cells). Scale bar: 10 cm. **d** Diagram of the different reference frames leading to four cell categories. Place cells and social-place cells respectively represent the imaged (blue) or the other (orange) mouse relative to the environment. Allocentric and egocentric social-vector cells represent the position of the other mouse relative to that of the imaged mouse, with coordinates centered at the head of the imaged mouse, rotating with it (egoSVC) or fixed to the arena (alloSVC). **e** Representative examples of the four cell types (columns). Each subpanel shows spatial maps (left) and calcium event plots (right) for one cell. Peak event rate (P) and spatial information (I) in the corresponding

reference frame is indicated. Scale bars: 20 cm. **f** Representative example of the spatial distribution of cells of each type in the field of view (color coded). **g** Left: Venn diagram representing the distribution of cells across categories. Right: Similar information in terms of percentage of cells in different combinations of categories. **h**–**i** Density of field peaks (left axis; bars) and coverage (right axis; line, mean ± s.e.m.) inside rings of increasing radii around the imaged mouse normalized by the area of the ring for alloSVCs (**h**) and egoSVCs (**i**). Note that around 70 cm, the maximum possible distance between mice, field density and coverage decay because conditions for animals to be at exactly 70 cm from each other are rarely met. **j** Cross-validated decoding error in each reference frame compared with shuffling (one dot per session; mean ± s.e.m., two-tailed Mann–Whitney test p-value indicated. From left to right, session number: 321, 182, 295, 338; Mann–Whitney $U$: 1779716, 1137944, 2432712, 3255215; Cliff's Delta: −0.65, −0.31, −0.44, −0.43). Source data are provided as a Source Data file.

and Supplementary Fig. 3b; cutoff value for classification: 95th percentile of the shuffled distribution for information). The classification was similar for a variety of modifications in selection criteria (Supplementary Figs. 4–5).

We found no signs of topographical organization of cell categories along the field of view (Fig. 1f and Supplementary Fig. 3c). The category with the highest percentage of cells was selfPC (42.7%), followed by egoSVC (13.1%) and alloSVC (11.1%, but see Supplementary Fig. 6) (Fig. 1g). In contrast, only a marginal number of cells was classified as socialPC (2.3%), suggesting that during free interactions social-vector coding is a preferred strategy among CA1 neurons. The density of peaks of social-vector fields was homogeneous across most of the effective arena and only decayed close to the edges, where coverage also decreased (Fig. 1h–i). In contrast, socialPCs were found to peak more often close to the borders, a characteristic previously described in selfPCs (Supplementary Fig. 3d–g).

We next asked whether cells were grouped into mutually exclusive subpopulations or instead expressed conjunctive coding. The percentage of cells within a category that were not conjunctive was higher for selfPC (69.1%) and egoSVC (45.1%), and lower for alloSVC (18.8%) and socialPC (22.0%) (Fig. 1g). We used an overlap index to assess whether pairs of cell categories tended to group the same cells or, in contrast, overlaps between them were of a random nature (see Methods). We observed that the overlap was high between alloSVC and other categories, as well as between selfPC and socialPC (Supplementary Fig. 3h). For the rest of cell type pairs, conjunctive coding occurred at a rate expected by chance, indicative of independence between mechanisms of cell specialization. In particular, chance-level overlap between the selfPC and egoSVC subpopulations made it unlikely that the firing of the latter could be explained by the position of the observer relative to the arena. Furthermore, percentages of cell types did not vary substantially after removing episodes during which the movement of both animals was correlated (Supplementary Fig. 4a). In contrast, the high overlap between selfPC and alloSVC, together with geometrical considerations, suggests that a fraction of cells classified as alloSVC are false positives (Supplementary Fig. 6). The observations above led us to focus for the remaining part of this work on egoSVC rather than alloSVC, because they represented both an overall larger and more independent subpopulation.

Our results point to relative rather than absolute position as a prevailing CA1 strategy to represent conspecifics at the single cell level in mice. To understand if this is also the case at the population level, we implemented a naïve Bayesian decoder[22] to predict the position of the self or the other in each reference frame based on the activity of cells belonging to the corresponding category (Fig. 1j and Supplementary Fig. 7). To estimate baseline accuracy levels, we trained the same decoder with shuffled data (100 shuffles for each session). We found that session-averaged cross-validated decoding error was lower than baseline for all cell categories, although the size and significance of the difference was higher for selfPCs, followed in order by egoSVCs, alloSVCs and socialPCs (note that higher absolute observed and shuffled errors for social-vector cells relate to a larger effective arena; Supplementary Fig. 3a).

Collectively, these results suggest that CA1 implements a multiplexed representation of the position of conspecifics during free interactions, favoring social-vector coding over other strategies. Among social-vector cells, egoSVCs stand out as a larger and more specialized population.

### Social-vector maps are modulated by identity

We next asked whether or not social spatial representations depend on the identity of the conspecific. To answer this question, we imaged from a mouse free foraging in an open field arena together with two cage mates (Fig. 2a). To differentiate them, we attached a LED to the head of one of them (mouse 1). We found that a substantial number of egoSVCs only represented the relative position of one mouse (Fig. 2b–d), with similar levels of information rate for the subpopulations of cells specialized in each one (Fig. 2e). The overlap index between these two subpopulations indicated that the conjunctive representation of both animals occurred at a rate higher than expected by chance, hinting to the possibility of some level of generalization across animals. To characterize it, we divided every session in two halves, obtained maps for each half and assessed their similarity through the Pearson correlation. The maps for the two halves could be constructed using coordinates of either the same or the other conspecific sharing the arena together with the imaged mouse. We observed that the correlation was higher when both social-vector maps of a given cell referred to the same mouse, indicating that egoSVCs encode mouse identity (Fig. 2f). However, we also found that the correlation between maps referring to different mice was higher for cells that were classified as egoSVC for both mice than for cells specialized in one mouse. This observation suggests incomplete remapping, or some degree of generalization across individuals in social-vector representations.

We next asked if similar observations could be made at the population level. To address this question, we trained a decoder with the relative position of one mouse and used it to decode the relative position of either the same or the other mouse, using all cells classified as egoSVC for at least one of them (Fig. 2g). We observed that both decoders performed better than one trained with shuffled data, and that same-mouse decoding produced lower errors than cross-mouse decoding. These results, compatible with observations at the single-cell level, suggest that egoSVCs represent the identity of conspecifics, while exhibiting some limited level of generalization across individuals. Similar conclusions were obtained when performing these analyses on alloSVCs (Supplementary Fig. 8).

### Modulation by familiarity

Since social interactions are modulated by familiarity, we also asked if social representations were similar for cage mates vs. other conspecifics. For up to 10 consecutive days, we trained a mouse to free forage in an open field environment in two sessions, the first one sharing the arena with a cage mate and the second one with an unfamiliar mouse (Fig. 3a). Familiar sessions came always first to avoid loss of interest primed by prior interaction with the unfamiliar mouse. Even with this bias, overall interaction time (including chasing, mounting, etc.) was significantly higher for the unfamiliar mouse, with no signs of adaptation after 10 days of daily exposure (Fig. 3b–c and Supplementary Fig. 9a, b).

To understand if this behavioral difference was associated with distinct egoSVC representations, we next compared egoSVCs specialized in the familiar vs. the unfamiliar mouse (Fig. 3d and Supplementary Fig. 9c). We observed that both subpopulations represented a similar proportion of cells, with overlaps higher than expected by chance and evidence for some level of generalization across individuals (Fig. 3e–g). This is compatible with our findings in the three-mouse experiment. We also found that information content was slightly higher for familiar egoSVCs (Fig. 3h), suggesting that coding of conspecifics could be influenced by prior experience and does not improve with the amount of interaction within the environment. Similar analyses on alloSVCs exhibited the same trend but were borderline on the significance of modulation by familiarity (Supplementary Fig. 9c–j).

We next asked if the increased information content for familiar mice allowed for better decoding at the population level when all egoSVCs were used (Fig. 3i). We observed that, in contrast with our expectations, decoding error was slightly lower for the unfamiliar mouse. To understand this apparent contradiction between information content of individual cells and population decoding, we examined

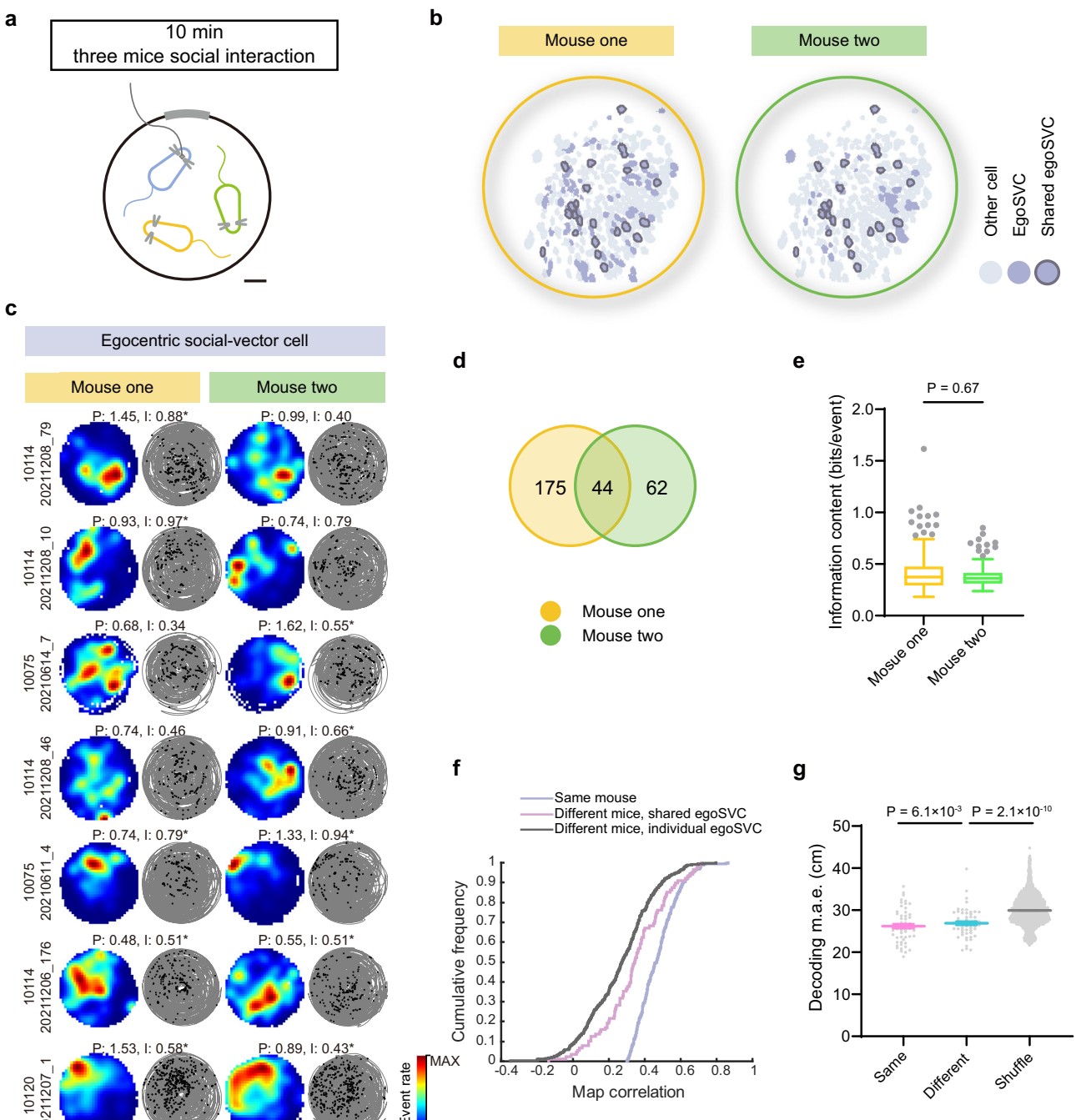

**Fig. 2 | Egocentric social-vector cells represent mouse identity. a** Diagram of the experimental paradigm. An imaged mouse (blue) interacted freely with two social partners (orange and green). Scale bar: 10 cm. **b** Representative example of the spatial organization of cells classified as egoSVC (dark purple) for each mouse and for both (dark edge). **c** Representative examples (organized as in Fig. 1e) of cells classified as egoSVC (asterisk) for one or both mice. Scale bar: 20 cm. **d** Venn diagram of cells classified as egoSVC for each mouse (281 cells out of 897 recorded from 7 mice; overlap index: 1.7, two-tailed Binomial test, $p = 8.9 \times 10^{-4}$).
**e** Distribution across sessions of mean spatial information content of cells classified as egoSVC for each mouse ($n$ (mouse one) = 219 sessions, $n$ (mouse two) = 106 sessions, median ± i.q.r., two-tailed Mann–Whitney test, Mann–Whitney $U = 11267$, Cliff's Delta = 0.03, $p = 0.67$). **f** Cumulative distribution of correlation between maps corresponding to the first and second halves of a session. For each half of the session, maps were constructed using the relative position of the same

(blue) or a different mouse (pink and gray), for cells that were classified as egoSVC for both mice (pink) or only one mouse (gray) (Kruskal–Wallis test, H = 254.1, $p = 6.8 \times 10^{-56}$; Dunn's multiple comparisons test, all $p < 0.002$). **g** Cross-validated error obtained when decoding the relative position of a mouse with a decoder trained with data corresponding to the same (pink) or the other (blue) mouse, together with the shuffled distribution (gray) (one dot per session, mean ± s.e.m., 55 sessions; two-tailed Wilcoxon matched-pairs signed rank tests for same vs. different mice: W = 648, Cliff's Delta = −0.12, $p = 6.1 \times 10^{-3}$; two-tailed Mann–Whitney test for different mice vs. shuffle: Mann–Whitney $U = 76096$, Cliff's Delta = −0.50, $p = 2.1 \times 10^{-10}$). In box plots, the central line indicates the median, and the bottom and top edges of the box mark the interquartile range. Whiskers extend from −1.5 × i.q.r. to +1.5 × i.q.r. from the closest quartile, where i.q.r. is the interquartile range. Black dots mark outliers. Source data are provided as a Source Data file.

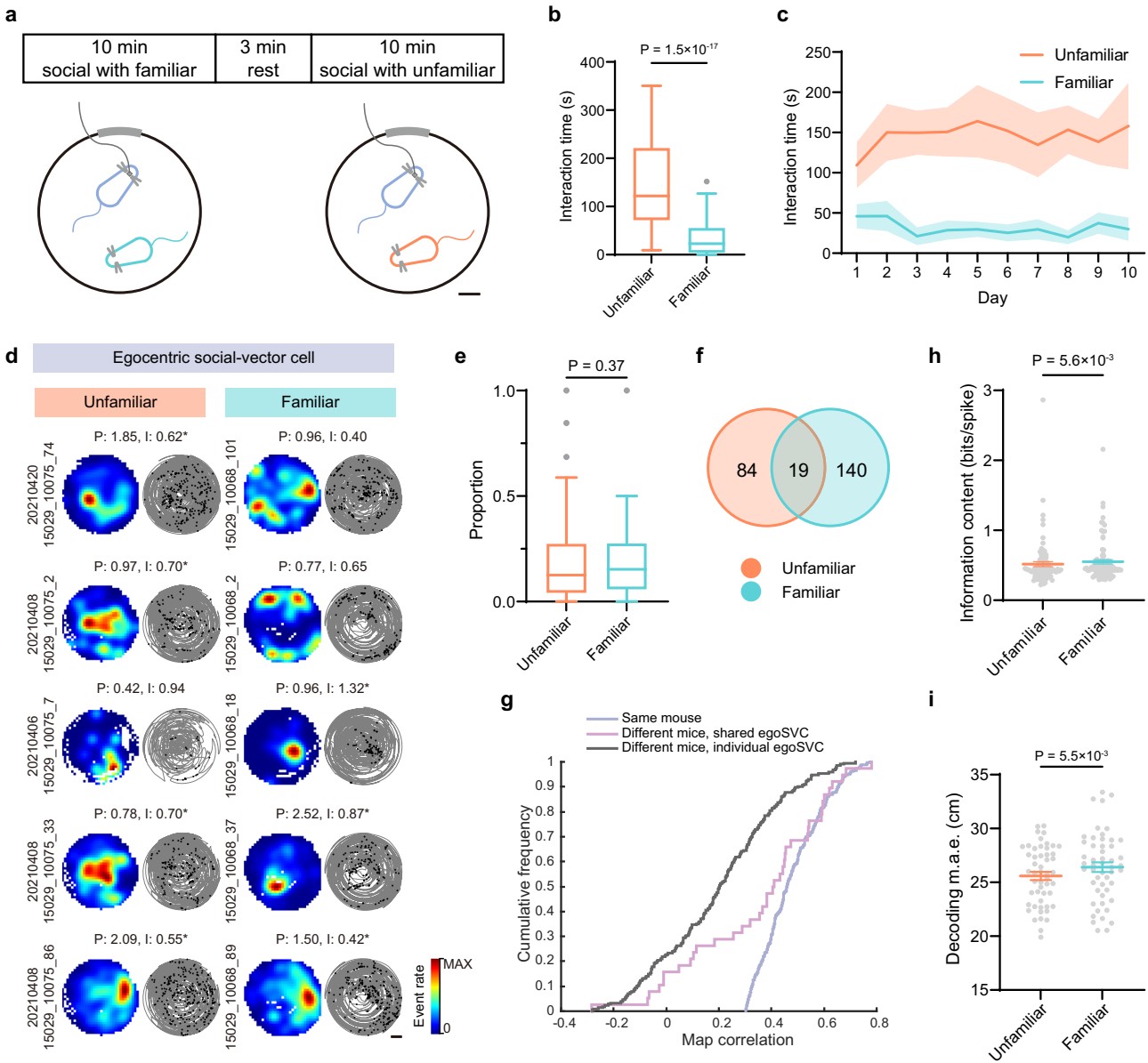

**Fig. 3 | EgoSVC representation of familiar and unfamiliar conspecifics.**
**a** Diagram of the experimental paradigm. An imaged mouse (blue) interacted freely in two consecutive sessions with either a familiar (light-blue) or an unfamiliar (orange) partner. Scale bar: 10 cm. **b** Time spent in social interactions with the familiar or unfamiliar conspecific ($n = 69$ paired sessions; median ± i.q.r.; two-tailed Wilcoxon matched-pairs signed rank test, W = −2345, Cliff's Delta = 0.81, $p = 1.5 \times 10^{-17}$). **c** Time spent in social interaction across 10 days of experiments (mean ± s.e.m.). **d** Representative examples (plots as in Fig. 1e) of cells classified as egoSVC (asterisk) during unfamiliar (left) or familiar (right) sessions. Scale bar: 20 cm. **e** Proportion of egoSVCs during unfamiliar and familiar sessions ($n = 106$ paired sessions; median ± i.q.r.; two-tailed Wilcoxon matched-pairs signed rank test, W = 494, Cliff's Delta = −0.08, $p = 0.37$). **f** Venn diagram representing cell subpopulations (8 imaged mice, 243 out of 1948 imaged cells; overlap index: 2.26, Binomial test, $p = 0.0013$). **g** As in Fig. 2f, cumulative distribution of the correlation

between maps corresponding to the first and second halves of a session describing the position of the same (blue) or a different mouse (pink and gray), for cells that were classified as egoSVC for both mice (pink) or only one mouse (gray) (Kruskal–Wallis test, $H = 174.9$, $p = 1.0 \times 10^{-38}$. Dunn's multiple comparisons test, all $p < 0.015$). **h** Distribution across sessions of mean spatial information content for unfamiliar and familiar egoSVCs ($n = 88$ paired sessions; mean ± s.e.m.; Two-tailed Wilcoxon matched-pairs signed rank test, W = 1322, Cliff's Delta = −0.10, $p = 5.6 \times 10^{-3}$). **i** Cross-validated errors when decoding the relative position of unfamiliar or familiar mice (mean ± s.e.m., 51 paired sessions; Two-tailed Wilcoxon matched-pairs signed rank test, W = 1021, Cliff's Delta = −0.14, $p = 5.5 \times 10^{-3}$). In box plots, the central line indicates the median, and the bottom and top edges of the box mark the interquartile range. Whiskers extend from −1.5 × i.q.r. to +1.5 × i.q.r. from the closest quartile, where i.q.r. is the interquartile range. Black dots mark outliers. Source data are provided as a Source Data file.

the error within sessions as a function of different key variables: familiarity (whether the session included a familiar or unfamiliar mouse), interaction (whether or not at a given point in time animals were interacting) and distance between mice. We found that, rather than familiarity or interaction, the difference in decoding error was explained by the higher amount of time spent at a close distance, which in the egoSVC reference frame is biased toward smaller errors (Supplementary Fig. 10).

Together, these results suggest that although mice interact more with unfamiliar conspecifics, egoSVCs represent more accurately the position of familiar ones, although this difference is small.

**Remapping of social-vector representations across contexts**
Given that hippocampal place cells remap across environments, we also asked if context modulates the social representation of egoSVCs. We imaged from a mouse while it foraged freely

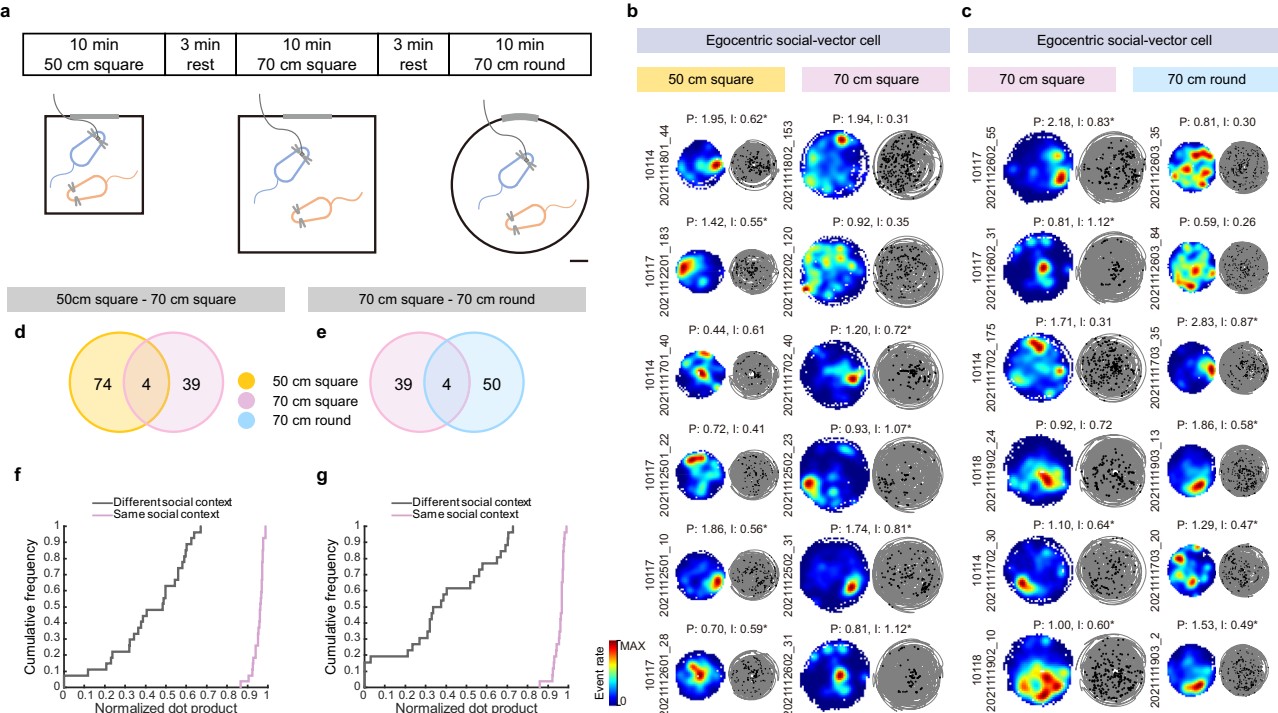

**Fig. 4 | Egocentric social vector representations are context dependent.**
**a** Diagram of the experimental paradigm. An imaged mouse (blue) interacted freely with a partner in three different contexts. Scale bar: 10 cm. Representative examples (as in Fig. 1e) of cells classified as egoSVC (asterisk) in at least one of two consecutive sessions, testing for effects of size (**b**) or shape (**c**). Scale bar: 20 cm. **d**, **e** Venn diagrams representing overlaps between subpopulations of cells classified as egoSVC in consecutive sessions (4 imaged mice, 1003 imaged cells; overlap indexes sessions 1 vs. 2: 1.20, Binomial test $p = 0.58$; sessions 2 vs. 3: 1.73, Binomial

test $p = 0.30$). **f**, **g** Cumulative distribution of the normalized dot product, quantifying the similarity of the mean firing rate of egoSVCs across halves in the same context (violet) or across contexts (gray) of different size (**f**) or shape (**g**). Halves of sessions in the same context were more similar than in different contexts (27 sets of paired sessions, two-tailed Mann–Whitney test, sessions 1 vs. 2: Mann–Whitney $U = 0$, Cliff's Delta $= 1$, $p = 10^{-15}$; sessions 2 vs. 3: Mann–Whitney $U = 0$, Cliff's Delta $= 1$, $p = 4 \times 10^{-15}$). Source data are provided as a Source Data file.

accompanied by the same conspecific during three consecutive sessions in different familiar environments: a small 50 cm side square box, a large 70 cm side square box and a 70 cm diameter circular box (Fig. 4a). Comparisons between consecutive sessions thus addressed the questions of environment size (1 vs. 2) and shape (2 vs. 3). We found that both types of contextual change produced remapping in social-vector representations, so that many cells were classified as egoSVCs only in one of the environments (Fig. 4b, c and Supplementary Fig. 11a). Furthermore, overlap indexes were not significantly higher than 1, suggesting that, unlike the modest generalization across individuals we found previously, egoSVCs exhibit no generalization across contexts (Fig. 4d, e). Since maps were different in shape and size, we could not assess their similarity through spatial correlation. Instead, we compared the activity of a cell across sessions using the dot product of the mean event rate[23]. We found that dot products between halves of the same context were significantly higher than those between halves of different contexts (Fig. 4f, g). The analysis of alloSVCs showed very similar results (Supplementary Fig. 11).

Collectively, these results suggest that CA1 social-vector representations are modulated by context, identity and familiarity of the conspecific sharing the behavioral arena. Importantly, this characterization contrasts with that of object-vector cells, which tend to generalize across objects and environments[13].

### Learning of a pursuit task improves social-vector coding

We next asked if social-vector representations are associated to performance in social tasks. To answer this question, we designed a behavioral paradigm in which a mouse learned to closely follow a conspecific (different across days) in a circular track to obtain a reward

(Fig. 5a and Supplementary Fig. 12). It took a variable number of sessions for mice to understand the aim of the task and associate it to the reward, after which the latency to reach the target decreased sharply (Fig. 5b, c). This allowed the classification of sessions as naïve (all initial sessions with a given imaged mouse) and trained (all sessions after the latency dropped to less than 30% relative to the previous day).

Imaging throughout the learning process allowed us to compare egoSVC representations in naïve and trained mice (Fig. 5d). We observed that learning did not affect the proportion of cells classified as egoSVCs or their mean calcium event rate. However, egoSVCs in trained mice had higher levels of spatial information content, related to both a lower number of fields and a smaller field size, hinting to a role of plasticity in sharpening their place fields through competitive learning (Fig. 5e and Supplementary Fig. 13b-c).

We next asked if these differences had an impact at the population level, affecting the capacity to decode the position of the chased mouse. We observed that cross-validated decoding errors were smaller for trained animals than for naïve ones (Fig. 5f). These results suggest that egoSVC representations improve when animals are engaged in a social task. We found that most of these conclusions apply also to alloSVCs (Supplementary Fig. 14).

Next, we asked what kind of changes in egoSVC representations took place due to learning. The geometry of the task, with head direction mostly aligned to the track (Supplementary Fig. 13a), allowed us to examine the activity of egoSVCs along a single dimension, represented by the angle α, measuring the relative position from the center of the annular track (Fig. 6a). For each cell, we constructed two maps of activity as a function of the angle, one for clockwise trajectories of the imaged mouse ($-360° \le α \le 0°$, conspecific on the right side) and one for counter-clockwise trajectories ($0° \le α \le 360°$,

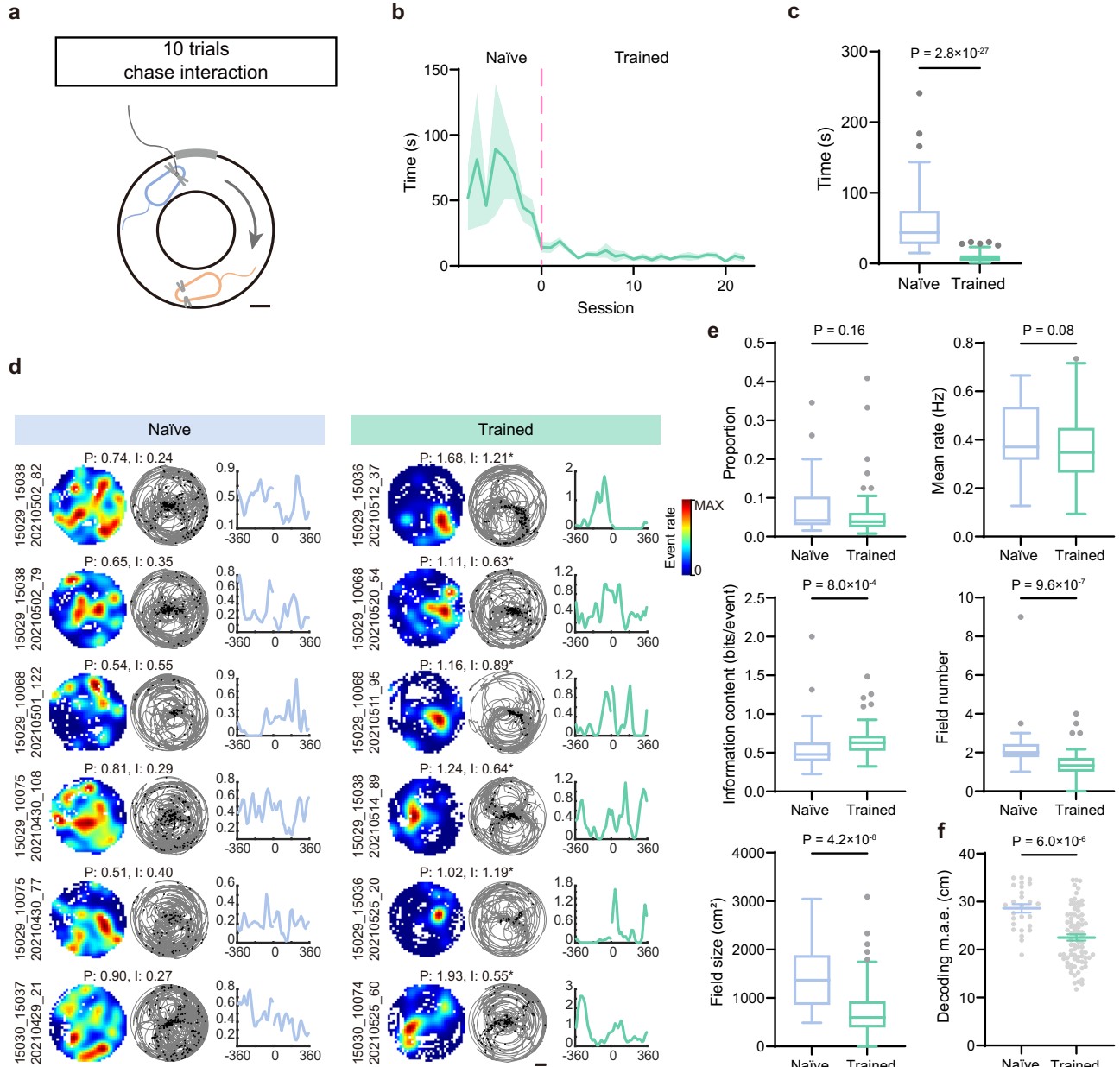

**Fig. 5 | Social vector representations are modified by training in a pursuit task.**
**a** Experimental paradigm of the pursuit test in a circular track. Scale bar: 10 cm.
**b** Distribution of the latency to obtain a reward across days (5 imaged mice; mean ± s.e.m.), centered around the transition between naïve and trained (day 0).
**c** Distribution of latency for naïve and trained sessions (n (naïve) = 35 sessions, n (trained) = 117 sessions; median ± i.q.r., Mann–Whitney test, two tailed, Mann–Whitney U = 73, Cliff's Delta = 0.96, p = 2.8 × 10⁻²⁷). **d** Representative examples of cells classified as egoSVC in trained (right) but not in naïve (left) sessions (one cell per row). Each example includes the rate map (left), the trajectory with events (center) and the rate vs. angle map. Maximum rate (P) and information content (I) are indicated. Scale bar: 20 cm. **e** Subpanels show for naïve (blue; n = 27 sessions) and trained (green; n = 81 sessions) conditions: proportion of cells classified as egoSVC, mean calcium event rate, mean information content, number of fields and field size (median ± i.q.r., p-values for two-tailed Mann–Whitney tests indicated, Mann–Whitney U = 897, 860, 637, 447.5, 370.5; Cliff's Delta = 0.18, 0.22, −0.42, 0.60, 0.67). **f** Cross-validated absolute decoding error for naïve and trained sessions (n (naïve) = 26 sessions, n (trained) = 79 sessions, mean ± s.e.m., two-tailed Mann–Whitney test, Mann–Whitney U = 440, Cliff's Delta = 0.57, p = 6.0 × 10⁻⁶). In box plots, the central line indicates the median, and the bottom and top edges of the box mark the interquartile range. Whiskers extend from −1.5 × i.q.r. to +1.5 × i.q.r. from the closest quartile, where i.q.r. is the interquartile range. Black dots mark outliers. Source data are provided as a Source Data file.

conspecific on the left side), where values close to 0° indicate that the conspecific is right in front of the imaged mouse and absolute values close to 360° indicate that the conspecific is right behind (Fig. 5d). We compared the collection of angle maps obtained for naïve and trained mice, sorting them according to the value of α where the maximum calcium event rate was observed (Fig. 6b). For both conditions we observed a clear diagonal band, which indicated on one hand that all angles are homogeneously represented and on the other

that cells tend to specialize in clockwise or counterclockwise trajectories, consistent with observations in the open field (Supplementary Fig. 3e, g). However, the band looked sharper in trained animals. To quantify this, we obtained the information content for angle maps, and observed that it was higher for trained animals (Fig. 6c). We also computed the population autocorrelogram by correlating the collection of maps (population vector) with rotations toward the front or the back of the imaged mouse (Fig. 6d). We observed that the population

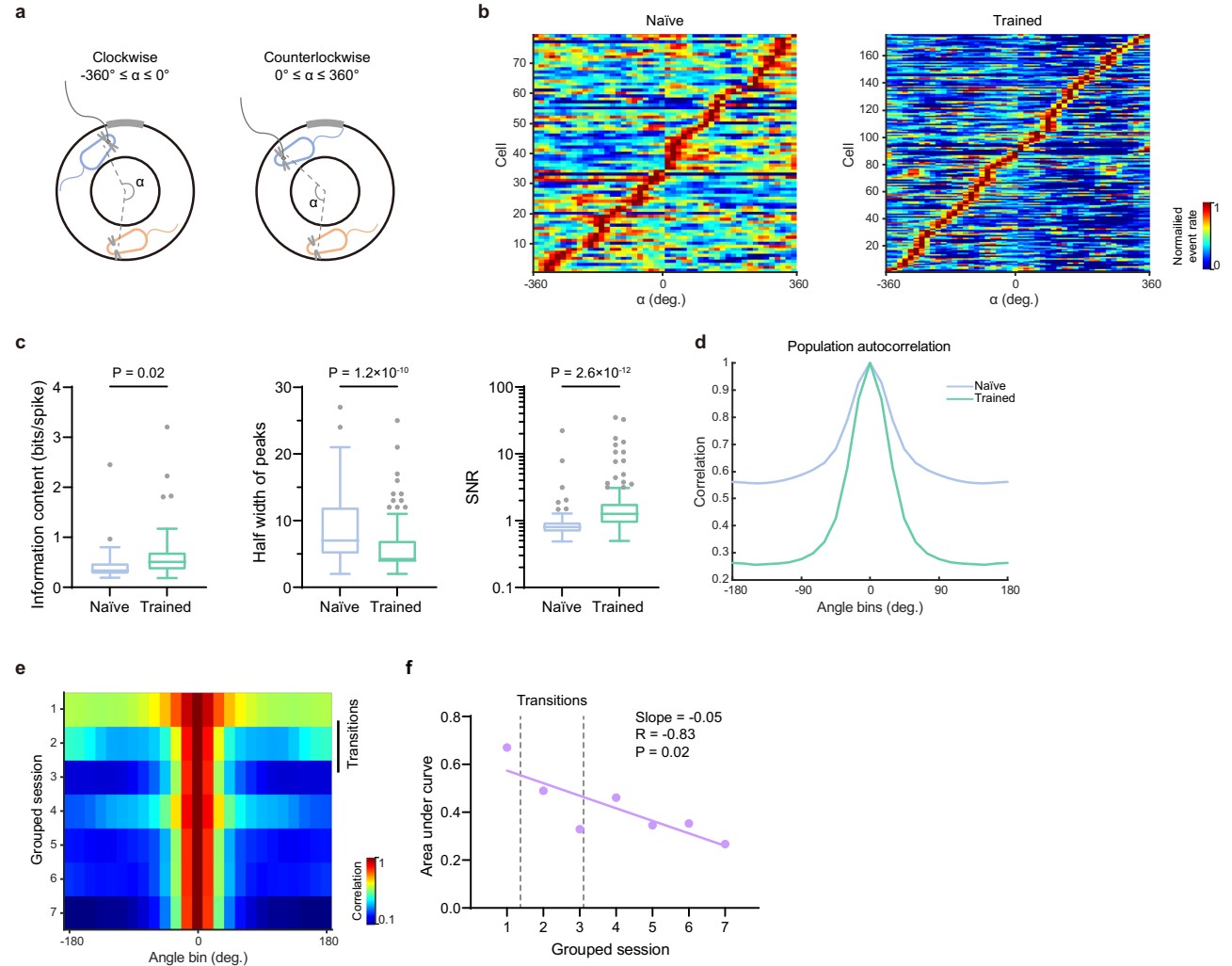

**Fig. 6 | Sharpening of egoSVC fields with training revealed by angle rate maps.** **a** Schematics of the angle α for clockwise and counter-clockwise trajectories. **b** Angle rate maps for all egoSVCs in naïve (left) and trained (right) sessions, ordered according to the angle of peak activity. **c** Distribution across sessions of mean information content (left, n (naïve) = 23 sessions, n (trained) = 71 sessions), and across cells of peak half-width (center, n (naïve) = 79 cells, n (trained) = 175 cells) and signal-to-noise activity ratio (right, n (naïve) = 79 cells, n (trained) = 175 cells) in both groups of angle maps (median ± i.q.r., two-tailed Mann–Whitney test, Mann–Whitney U = 546, 3494, 3120, Cliff's Delta = −0.33, 0.49, −0.55, p = 0.02, $1.2 \times 10^{-10}$, $2.6 \times 10^{-12}$). **d** Population autocorrelation for the pool of all naïve (blue) and trained (green) maps (difference in area under the curve: 0.26, p-value through

direct comparison with 1000 shuffles of the naïve/trained tags: $p < 0.001$). **e** Evolution of the population autocorrelogram, for groups of 5 consecutive sessions pooled together. Black line: Sessions including transitions between naïve and trained. **f** For data in **e** evolution of the area under the autocorrelogram curve (two-tailed Pearson correlation test, R = −0.83, p = 0.02). Dashed lines: transitions between naïve and trained. In box plots, the central line indicates the median, and the bottom and top edges of the box mark the interquartile range. Whiskers extend from −1.5 × i.q.r. to +1.5 × i.q.r. from the closest quartile, where i.q.r. is the interquartile range. Black dots mark outliers. Source data are provided as a Source Data file.

vector autocorrelation for trained animals exhibited a sharper decrease to overall lower levels. In consequence, the difference in area under the autocorrelation curve was significantly higher than expected by chance, as represented by the distribution obtained from shuffling the naïve/trained label of cells. We also defined for each map a signal region, including the peak event rate bin and the collection of bins around it defined by half-decay in event rate, and the complementary noise region. We observed that the signal region was smaller for trained animals, although it concentrated a higher fraction of the overall cell activity (Fig. 6c). These observations suggest that as animals learn the task egoSVCs become sharper and more selective.

To visualize the learning dynamics associated to the task, we pooled all maps and tagged them according to the experience of the animal in the task (sessions 1 to more than 30). We then obtained the autocorrelogram for all cells in non-overlapping groups of 5 consecutive sessions (Fig. 6e). A progressive reduction in the width of

the autocorrelation was observed, even after all transitions from naïve to trained had taken place. We quantified this overall reduction using the normalized area under the curve for each group of sessions, which exhibited a significantly negative correlation with experience (Fig. 6f).

Put together, these results suggest that plasticity plays a major role in shaping social vector representations in CA1 when animals learn a social task. While the number of egoSVCs does not change and fields remain homogeneously distributed across space, they become sharper, more specialized on a single location of the effective space, and allow for a better decoding of the relative position of the other.

## Discussion
In this study, we examined social-spatial coding in the hippocampus during free or trained social behaviors. Our main result is that the representation of others in CA1 is multiplexed, but mostly done through egocentric social-vector coding, implying that neurons code

the position of a conspecific relative to self-position. Previous studies in rats and bats examined this question in experiments where a passive observer watched a conspecific do a task for imitation purposes[10,11]. This kind of setup allowed for the groundbreaking discovery that CA1 is involved in social-spatial representations. However, the ambiguity intrinsic to the paradigm did not allow to dissect whether this coding used the environment or self-position as a reference frame, since the observer had only marginal displacements relative to the environment. Our experiments, while fully compatible with these results, provide evidence against the interpretation of a purely allocentric coding of others in CA1. We show that in freely interacting mice, the position of the other relative to the environment is coded by a marginal population of cells. Instead, a greater subpopulation of cells represents the relative position of conspecifics in either allocentric or egocentric coordinates. These results point to a strong asymmetry in the way in which the hippocampus treats the self and others, differentiating it from a mirror-like system[24], although potential differences between species in this aspect of CA1 coding require further assessment.

While previous results have shown that the collective dynamics of CA1 can simultaneously represent the position in space of multiple bodies (e.g., the self and a conspecific), a less explored idea is whether or not it can represent a single external body from multiple perspectives. Our results indicate that, although not all representations are equally predominant, this is indeed the case. During free social interactions, the position of conspecifics is represented relative to self-position in at least two coordinate systems: one in which the direction of the axis is fixed to the environment (allocentric social-vector cells) and one that rotates with self-head direction (egocentric social-vector cells). The first group, which proved hard to assess from free exploration experiments given a bias toward false positive classification from place cells firing close to the border of the environment, could be related to object-coding cells in the medial entorhinal cortex[13]. The second, and the widely dominant one in terms of number of dedicated cells, recalls egocentric coding of objects in the lateral entorhinal cortex[14]. It is possible that diverse perspectives of the relative location of a single conspecific converge into the hippocampus through these separate pathways.

The hippocampus is known to encode non-spatial information in a spatial context[4,25–27]. For example, a hippocampal cell can be activated by the combination of a specific odor at a preferred location within a given environment[28]. During social behavior, animals receive spatial and non-spatial cues from conspecifics, many of which are relevant to assess potential interactions. Our findings indicate that social-spatial representations remap across conspecifics and environments. Similar to what has been found for other types of non-spatial information, this is evidence supporting a combinatorial strategy to deal with the convergence of multiple independent sources of information. Somewhat contrasting this general notion, we found some traces of a limited degree of generalization across individuals. We also found that the position of familiar conspecifics is slightly more accurately coded in CA1 than that of unfamiliar ones, pointing to a modulation by accumulated prior experience in other environments. The processing pathway of this information could involve CA2, which has been shown to code familiarity with low spatial specificity[29]. This modest level of generalization across individuals and lack of generalization across contexts differentiates social-vector coding from object-vector responses found in the entorhinal cortex and to a lower degree in the hippocampus[6,8,9,12,13].

Social-spatial representations were originally characterized in a trained behavior[10,11]. Our findings of analogous coding during spontaneous behavior raise the question of whether learning of a social task only utilizes or instead modifies pre-existent CA1 representations. To address this point, we trained mice in a new pursuit task where a clear-cut division between trained and naïve animals was possible. Our results show that learning to closely follow a conspecific modifies

representations, improving social-spatial coding. This is achieved by improving the tuning of individual social-vector cells rather than by recruiting a greater number of them. This result goes in the same direction of our observation of improved social-spatial coding of familiar relative to unfamiliar conspecifics at the single cell level, hinting to the possibility of a common mechanism underlying both phenomena.

Taken together, our results show that, in a social setting, multiple representations of self and others coexist in the hippocampus, which could perhaps maximize available information about all potential interactions within an environment. Whether or not this multiplexed strategy responds to different pathways of information reaching the hippocampus needs to be clarified by exploring upstream social-spatial coding, both within and outside of the hippocampus.

## Methods
### Mice
All procedures for animals were approved by Animal Care & Use Committees at Peking University. A total number of 48 C57BL/6 N mice (about 12 weeks old; including imaged and not imaged) were used. In order to minimize behaviors related to sex or dominance, all mice were female. All mice were group-housed with a 12h-12h light-dark cycle in a temperature controlled (20 °C), 50% humidity and air-circulating cabinet and provided with food and water as libitum. Behavioral experiments were conducted during the dark phase.

### Surgery
**Viral injection.** Mice were anaesthetized with isoflurane and placed in a stereotaxic instrument (KOPF). In each mouse, two injections of 200 nL of AAV2-1-Ef1a-GCaMP6s (titer: $7.6 \times 10^{12}$ vg/mL) were injected into the CA1 subfield of the hippocampus at a rate of 100 nL/min in two sets of injections. The two injection coordinates were −1.5 mm ML, −1.82 mm AP, −1.4 to −1.2 mm DV and −2.28 mm ML, −2.5 mm AP, −1.4 to −1.2 mm DV relative to bregma. Animals recovered for at least 2 weeks before GRIN lens implantation.

**GRIN lens implantation and fixation of baseplate.** After a 2 mm diameter craniotomy was opened around injection coordinates, and the dura, cortex and portion of corpus callosum were aspirated with saline, the 1.8 mm GRIN lens was implanted aimed directly to the dorsal hippocampus CA1 (Edmund Optics, 670 nm, 0 mm working distance). The GRIN lens was secured to the skull with glue and dental cement. After a 2 weeks recovery period, a third surgery was performed to position the baseplate optimally and cement it to the skull. A miniscope was used to find the position in which the largest number of cells appeared clear in the confocal plane.

### Behavior assay
**Free social interaction in the open field.** Two or three mice were simultaneously placed in a round open field (diameter 70 cm) for free social interaction in sessions lasting 10 min each. In all experiments mice were cage mates, with the exception of tests for the representation of unfamiliar conspecifics. During each session, behavior was filmed with a camera positioned in the lab ceiling. Calcium imaging videos were also obtained from one animal in each session using a miniscope. The miniscope was connected to a digital acquisition box (DAQ) through a long and flexible cable. Each animal was habituated to the open field and to wearing the miniscope for at least one week before social experiments. For two-mouse interactions, 12 imaged mice and 11 non-imaged mice were used, including 14 pairs of unfamiliar interactions and 14 pairs of familiar interactions. For three-mouse interactions, 7 groups of mice were used. Two control imaged animals (Supplementary Fig. 5a, b) were trained in two consecutive sessions with the same conspecific and in two consecutive sessions with an object that was displaced in between sessions.

**Social interaction across contexts.** An imaged mouse and its social partner were placed into a 50 × 50 cm square box, a 70 × 70 cm square box, and a 70 cm (diameter) round box in 3 consecutive 10-min sessions, with 5–10 min rest sessions in between. Four groups of mice were used.

**Pursuit training in the annular maze.** All animals were habituated to a reward during three days before training started. They were also familiarized individually with the arena, consisting of an annular maze (external diameter: 70 cm, internal diameter: 50 cm). In days prior to the social task, mice selected for the role of chasers (A) where familiarized with the presence of a reward every time a paperboard barrier was inserted inside the maze. In each session of the social task, mouse A was placed inside the annular maze and was left free to explore the maze for a variable number of seconds. After this, a second mouse (B) was place inside the maze at a random position. Whenever mouse A started following closely mouse B, the paperboard barrier was inserted, separating both mice. Mouse A went to find the reward while mouse B was removed from the maze, placed in a pedestal and given a reward. 10–15 trails were performed in each pair for each single day. During each session, calcium imaging videos from mouse A and behavior videos were simultaneously recorded and the time spent in each trial quantified. Mouse A was wearing a miniscope connected to a DAQ box through a long and flexible cable. In a few cases, mouse B was wearing a wireless miniscope, but the data collected from it was not used in this work. After each experiment, animals were fed ad-libitum for 6 h. Elapsed time to reach the target was used to classify mouse A as naïve or trained. All mice were initially naïve and they were classified as trained on session 0, defined as the first session in which median elapsed time fell below 30% of to the highest latency observed on the previous day. 5 mice were used as mouse A, and 9 mice were used as mouse B.

### Data analysis and statistics
Data were analyzed using MATLAB (2020b) and GraphPad Prism 9. Figures were plotted using MATLAB (2020b) and GraphPad Prism 9 and arranged in Adobe Illustrator 2022.

Statistics were performed by GraphPad Prism 9 and MATLAB, and for effect sizes we used function *meanEffectSize* from MATLAB.

### Calcium imaging signals
Calcium imaging data were recorded using a miniscope (UCLA V3) through Miniscope DAQ Software (https://github.com/Aharoni-Lab/Miniscope-DAQ-QT-Software/releases/tag/v1.10). In two animals we found signs of spread depression, in the form of a slowly evolving calcium wave. These animals and their data were excluded from this work.

To extract calcium traces for individual cells, an open-source analysis pipeline (https://github.com/etterguillaume/MiniscopeAnalysis) was applied. Raw videos from each session were first processed by using the NoRMCorre algorithm for motion correction[30] (https://github.com/flatironinstitute/NoRMCorre), followed by CNMFe (constrained non-negative matrix factorization for microendoscopic data)[31,32] (https://github.com/zhoupc/CNMF_E) to extract the raw calcium signals, deconvoluted calcium activity and the binarized neural activity for each of the cells. Calcium events were defined as time frames when the deconvolved calcium signal surpassed three standard deviations.

### Same cell registration across sessions
We used CellReg[33] (https://github.com/zivlab/CellReg) to identify the same cell across sessions. The alignment type of registration within the same day was 'Non-rigid', while the alignment type of registration across the day was 'Translations and Rotations'. The cells identified as being the same by CellReg were used for further analysis. Tracking of cell identity across sessions was used in Figs. 3 and 4.

### Position and social behavior
Behavioral videos obtained at 30 frames per second were analyzed based on the tracking of mouse body or miniscope LED using DeepLabCut[34,35] (https://github.com/DeepLabCut/DeepLabCut). DeepLabCut was also used to extract head direction from the location of two LEDs attached to the miniscope, that rotated with the head of the mouse. Positions that were out of the open field or incorrect were then removed and a 15 point mean filter was applied to smooth trajectories.

For social interaction experiments, behaviors were annotated manually frame by frame to identify the onset and offset time by using the Boris software[36]. Both social and non-social behaviors were annotated. Social behaviors included chasing, being chased, mounting, being mounted and other social interaction behaviors that involved mutual approach. Non-social behaviors included free moving, jumping and staying still. The level of social interaction was measured by using the percentage of time spent in social behaviors.

For pursuit experiments, we used the Boris software to annotate different periods of each trial, including free moving, chase and reward. The data from Boris was then processed by MATLAB using custom scripts.

### Rate maps, firing fields and spatial information
Codes for analyzing spatial maps were adapted from the Behavioral Neurology Toolbox, (c) Vadim Frolov 2018 (https://bitbucket.org/cnc-ntnu/bnt/src/master).

For each cell, we constructed four spatial maps using i) the position of the imaged mouse relative to the environment (selfPC), ii) the position of a conspecific relative to the environment (socialPC), iii) the position of a conspecific relative to the imaged mouse with coordinates that did not rotate (alloSVC) and iv) the position of a conspecific relative to the imaged mouse with coordinates that rotated together with the head of the imaged mouse (egoSVC).

To obtain event rate maps, the calcium transients and position data were sorted into bins of 2 × 2 cm. For social vector maps in the effective arena (Supplementary Fig. 3a) we used bins of 4 × 4 cm. The event rate in each bin was calculated by dividing the number of calcium transients by the amount of time spent in the bin. A 2D Gaussian kernel with standard deviation of 2 bins was used to smooth the rate map. The mean rate of each cell was calculated as the total number of calcium transients divided by the total duration of the session.

Place fields were detected as peaks in the smoothed event rate map higher than 0.5 Hz. The area of the place field was defined by contiguous pixels around the peak with an event rate above 60% of the peak value.

For each cell, the spatial information content[37] was calculated as

$$I = \sum_i p_i \frac{\lambda_i}{\lambda} \log_2\left(\frac{\lambda_i}{\lambda}\right) \tag{1}$$

where $\lambda_i$ is the mean event rate in the $i^{th}$ bin, $\lambda$ is the overall mean event rate, and $p_i$ is the amount of time spent by the animal in the $i^{th}$ bin normalized to 1.

To establish a cutoff value of information rate for a given cell and category, we obtained the corresponding information rate for 1000 shuffles. For each shuffle, the information rate was calculated after displacing the calcium trace in time and wrapping it around the total duration of the session. The cutoff value for cell classification was set as the 95th percentile of the shuffled distribution for each cell. Additionally, to be included in a cell category, cells had to have maps with a stability greater than 0.3. Stability was defined as the Pearson correlation between maps for the first and second half of the session.

### Overlap index
The overlap index between two cell subpopulations was defined as the number of cells belonging to both subpopulations divided by its expectation if the two subpopulations were independent. This

expectation was obtained as the overall number of cells multiplied by P, defined as the product of expectations of a cell falling into each category. Values of the overlap index significantly greater (lower) than 1 indicate an overlap higher (lower) than expected by chance. This was tested using the Binomial test with probability P.

## Position decoding

We used a Naïve Bayes decoder to decode the position of the self, the other, or the other relative to the self in the corresponding reference frame from neural activity[22]. All decoding was done using 5-fold cross validation. To do this, rather than randomly selecting data, sessions were divided into 5 consecutive segments, so that segment 1 corresponded to the initial part of the session and segment 5 to the final part. Sessions with no cells of the category being tested were discarded, which mainly influenced the number of available sessions in the socialPC category. We assumed that events from different cells followed independent Poisson processes. For each frame, the probability ($P$) of the animal being in a given position bin was calculated as

$$P(X|n) = CP(X)\left(\prod_{i=1}^{N} f_i(X)^{n_i}\right)e^{-\tau \sum_{i=1}^{N} f_i(X)} \quad (2)$$

where $X$ is the spatial bin, the element $n_i$ of vector $\mathbf{n} = (n_1, n_2, \ldots, n_N)$ is the number of calcium events for neuron $i$ within the time window, $C$ is a normalization factor, $P(X)$ indicates the overall probability of the animal occupying spatial bin $X$, $f_i(X)$ is the mean event rate of cell $i$ in spatial bin $X$, and $\tau$ is the length of time window. While the vector $n$ referred to testing data, other relevant variables referred to training data. The position with the maximum likelihood was taken as the decoded position of the animal on that frame. Shuffled decoding error was obtained from shuffled data, generated by displacing the position of animals in time and wrapping it around the length of the session. Decoding error for test data in each cross-validation step was defined as the mean absolute error (m.a.e.) between the true position and decoded position.

## Shuffling-in-place

To understand if selfPC activity could generate alloSVC or egoSVC false postives, we shuffled cell activity in a way that it did not affect selfPC maps. We first divided the physical arena in 10 cm wide bins. For each bin, we selected and stitched together the episodes during which the mouse occupied the bin. We shuffled the resulting array using a random shift that ensured that the shuffled array was at a distance of at least 5 s relative to the original one. For some bins with lower coverage this was not possible, and the shift resulting in the maximum distance was used. The 10 cm size of the bins was a compromise to reduce the number of such cases while not affecting shuffled selfPC maps. An instance of shuffling-in-place was the result for applying this procedure to all bins. After this, cell activity was reconstructed by re-placing all episodes in their original temporal order. After this, maps for all coordinates were calculated as in the rest of this work.

For classification of cells, only 10 shuffles-in-place were used, given that each new map required 1000 additional shuffles for proper classification. For comparisons of information content, 1000 shuffles were used for each cell.

## Cell activity across different environments

To compare the activity of cells across contexts of different size and shape, we used the normalized dot product of mean event rates[23]. We calculated the mean event rate of each cell in each condition divided by the maximal mean rate of that cell in all conditions. The values for each cell were grouped into a mean activity vector, with length equal to the number of cells. Vectors for each session were normalized by their Euclidean norm. To quantify similarity in cell activity across sessions, the dot product between the corresponding normalized vectors was obtained, which is equal to the cosine of the angle between them. To compare the activity of cells within or across sessions, all sessions were divided into halves and the dot product computed between first and second halves of either the same or different sessions.

## Angle maps

The geometry of egoSVCs in the annular maze allowed us to reduce maps to a single dimension. The position of two mice relative to the center of the annular maze was defined by two [X, Y] vectors, and α was defined as the angle between both vectors. This reduction assumes that animals mostly look to the front while navigating, while regular two-dimensional egoSVC maps do not make this assumption. To differentiate clockwise (conspecific to the right of the imaged mouse) from counterclockwise (conspecific to the left of the imaged mouse) trajectories, we introduced a negative sign to the former, which thus spanned from −360° to 0°. With this definition, angles close to 0° represented a situation in which the conspecific was closely ahead of the imaged mouse, while angles with absolute value close to 360° represented situations where the conspecific was close but to the back of the imaged mouse. For each cell, we constructed angle maps of cell activity as the number of calcium events divided by the time spent in each 15° wide angle bin. We used circular smoothing on each half of the map using a Gaussian kernel with standard deviation of 1 bin.

## Autocorrelogram

We used population autocorrelation to compare angle maps in naïve and trained animals. We correlated the collection of original maps with circularly shifted versions of them, from −180° to 180°. Each half of the angle map was shifted independently, but the correlation was obtained from both halves together. Positive (negative) shifts represented shifts toward the front (back) of the animal.

## Histology

After imaging and social interaction experiments, mice were transcardially perfused with saline and then 4% paraformaldehyde (PFA). After postfixed in PFA for 24 h, the brain was cryoprotected by 30% sucrose for another 24 h. 20-μm coronal section were obtained by using a cryostat (Leica).

## Immunostaining

Sections were rinsed by PBS and blocked by 2% Albumin Bovine – 0.3% Triton – PBS for 1 h. Then, slices were incubated with 1:500 chicken anti-GFP antibody (Invitrogen, A10262) overnight and followed by 1-h incubation with 1:500 Alexa Fluor 488 goat anti-chicken (Invitrogen, A11039) secondary antibody and DAPI. Slices were washed by PBS after incubation with antibodies. Finally, slices were mounted on slides and covered with mounting medium.

## Microscopy imaging

Images were acquired by Zeiss Axio Scan Z1 to confirm the position of lens implantation and the expression of GCaMP6s.

## Reporting summary

Further information on research design is available in the Nature Portfolio Reporting Summary linked to this article.

# Data availability

Data supporting this work are available in https://github.com/SherlockX-hub/SocialVectorCell-Public. Source data are provided with this paper.

# Code availability

Scripts supporting this work are available in https://github.com/SherlockX-hub/SocialVectorCell-Public.

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

## Acknowledgements

This work was supported by the National Key R&D Program of China (2019YFA0802400; C.M.), the Qidong-SLS Innovation Fund (2023002029 and 2020001540; C.M.), the Space Brain Project from the Lingang Laboratory, Grant No. LG-TKN-202204-01 (C.M.), Science Fund for Distinguished Young Scholars in Beijing (JQ23023; C.M.), Chinese Government Foreign Expert Project for Emilio Kropff (G2022101003L; C.M.), and PICT-2019-2596 grant from the Science Ministry of Argentina (E.K.). We thank the National Center for Protein Sciences at Peking University in Beijing, China, for assistance with imaging and histology.

## Author contributions

Q.C., E.K. and C.M. designed the research, X.Z., Q.C., K.G., A.L., C.C., S.C., Y.Z., R.L. and J.H., collected the data, X.Z., C.M. and E.K. analyzed the data, C.M. and Q.C. supervised the project, X.Z., E.K. and C.M. wrote the paper with input from all authors.

## Competing interests

The authors declare no competing interests.
