## [Peer Review File · Nature Communications]

REVIEWER COMMENTS

Reviewer #1 (Remarks to the Author):

The authors of this paper recoded social-spatial representations in CA1 area of the mouse hippocampus, in various behavioral tasks, and using calcium imaging. They report single-cell representations of social-place cells, allocentric social-vector cells, and egocentric social vector cells.

This is an important study which brings additional supporting evidence for new perspective on the mammalian hippocampal circuits – their involvement in social behavior.

The paper is well written, and the analysis is concise, yet I am not convinced by some of the results and the interpretations made by the authors in this paper.

Comments

1. Using the classical single cells tuning maps of place cells to untangle spatial representations of self, form spatial representation of others in CA1 is challenging. It is possible that correlated movement patterns between self and conspecifics may cause artifactual social-spatial representation. For example, a classical place cell representation could be misinterpreted to a social-spatial representation in an extreme case when the movement of the recorded mouse is correlated with the movement of conspecifics (for example when one mouse is chasing another mouse). The manuscript completely ignores this possibility, and no evidence is presented to rule this out. I suspect that the proportion of the different cell categories would drastically change after taking this into account, and this of course will directly impact the results and conclusions from the conjunctive coding analysis, the population analysis, etc. The authors need to add compelling evidence that their social place cells and social vector cells are not contaminated by self-spatial responses and cannot be explained by stereotypically correlated movements patterns.

2. I would also like to see some quantification of the movement patterns and coverage of the behavior in the social-vector cells. I suspect that biased coverage of head direction angles might also introduce artifactual social-vector representations.

3. The authors conclude and generalize in line 106 – that their results point to a relative rather than absolute position of social-spatial coding in CA1. This extrapolation to all mammalian CA1 hippocampus is not supported by their results. Previous studies on social place cells recorded from stationary observer while observing a moving conspecific, and from a different species (rats and bats). The most immediate and obvious possible explanation for the difference between their results and previous studies is a simple species-specific behavioral difference. This should be made explicitly in the text.

4. In relation to the social vector cells, I am missing quantification of the distribution of the distance-to-conspecific across social vector cells (ego and allo)- is it confined to small distance range or evenly distributed across all recorded distances?

5. In line 91 and in Figs. 1e, supp. 2a – it is not clearly explained what do the authors mean by “effective arena”, and why the “effective arena” is bigger for social vector cells?

6. In the pursuit tasks – the egoSVC is collapsed to a single angle on the track- if I understood correctly, there is an assumption that the head direction and the heading direction are similar in this task. If this is correct, then I would like to see some analysis of the head direction during pursuit -so as to see if this assumption holds true.

7. Fig. 6e is interpreted as sharpening the tuning – but to me it seems that instead of sharpening – the DC activity is lowered. If one would scale the tuning curve to min/max then it seems that the tuning is not sharpened but instead, there is a decrease in baseline of the tuning curve – which would mean an increase in SNR instead of sharpening of the tuning curve.

Reviewer #2 (Remarks to the Author):

The authors set out to understand the nature of the hippocampal representation of conspecific animals. It has been shown several times that hippocampal neurons respond to the position of not only the animal's current location but also the location of a conspecific. It is unclear, however, whether the representation of others is in a world-centered or self-centered reference frame because previous experiments recorded neural activity from a passive observer. Here, they allow female mice to freely interact or engage in a pursuit task. In addition to classical place cells (selfPC) and place cells representing the location of others (socialPC) they report two new classes of hippocampal CA1 neurons with vector-based responses to the conspecific in either allocentric or egocentric reference frames. In addition, they state that various factors, including mouse identity and familiarity, environmental context, and task modulate these responses.

This study asks an important and timely question of how animals track the location of others during social interactions. Recent technological advances in lightweight recording devices and quantitative behavioral tracking have made it possible to address such questions. The variety of experimental conditions and the ability to record large numbers of CA1 neurons across sessions using calcium imaging have yielded an impressive and valuable dataset. The figures are nicely presented with many example responses, and individual data points are used to show outcome variability clearly.

Major comments:

1. To claim a new functional cell type exists requires substantial evidence. Such evidence is lacking here. Cells were categorized into four non-mutually exclusive functional classes based on two criteria: spatial information content relative to a shuffled distribution and within-session map stability with a Pearson correlation > 0.3 . These are liberal criteria relative to what is commonly used in hippocampal literature. More rigorous criteria additionally include a minimum firing rate, a detectable firing field, and most importantly, stability between separate recording sessions. Between-session stability (usually assessed by Pearson correlation > 0.5) is critical to eliminate false positives due to biased behavioral sampling or confounding cues present in just one session. Without showing that these vectorial responses produce stable firing fields across different recording sessions, one cannot conclude these functional classes exist. Because the entire paper hinges on the existence of these new cell classes, this point must be addressed.

2. Related to the previous comment, sampling space in both allocentric and egocentric coordinates is difficult to achieve in short 10-minute sessions. This problem is exacerbated by the 5-minute within-session comparisons used to claim tuning stability. The sampling is especially problematic because of the potential interaction between the two reference frames. In other words, an animal may visit all places in the environment and all vectors relative to the conspecific but will rarely visit all combinations of the two (e.g., north of the conspecific while being in the south end of the arena). This is visible in some examples in Supplemental Figure 2. This biased sampling will necessarily confound the interpretation of maps using one reference frame while ignoring the other reference frames. This could be addressed by plotting the coverage of the space in several reference frames against each other. The authors should require that coverage and occupancy in the other three reference frames also meet some acceptable threshold before classifying the cell. Additionally, the authors must show maps in all reference frames for each cell that meets the criteria for one or more of the defined cell types.

3. Decoding of position is used in this study to justify that information about spatial location is present in different populations of cells. The decoding error is higher than expected in Figure 1J for traditional place cells (selfPC), where approximately half of the sessions are shown in this plot to have a decoding error comparable to the Shuffle group (ca. 20 cm). Why is the decoder so inaccurate using standard place cells? It would be insightful for the authors to investigate why this error rate is so high. Additionally, the decoding error is substantially worse for all other cell classes relative to their respective shuffled groups,

and significant differences between the true distribution and shuffled distribution are likely due to such a large sample size (see next comment).

4. The interpretation and reporting of statistical results, specifically regarding P-values, is flawed and could be improved by accounting for additional details like effect size. For example, Figure 1J uses P values from a Mann-Whitney test between two samples. These P values are extremely small, leading the authors to conclude there is a significant difference between the data and the shuffled groups, and therefore, these functional classes contain meaningful levels of information. This interpretation is flawed because it does not examine the effect size. Having small P values is completely meaningless on its own because if one collects enough samples the P value gets smaller by definition. This problem persists throughout the paper and undermines the majority of claims. Effect size must be reported everywhere, along with other statistical details, including sample size, degrees of freedom, and test statistics.

5. The authors fail to cite and discuss several relevant studies and thus do not place their work in appropriate context. Egocentric coding in the hippocampus has been reported in bats (Sarel 2017), mice (Jercog et al., 2019), and rats (Ormond et al., 2022). Egocentric coding similar to the type discussed here has additionally been found in several brain areas important for spatial navigation, including posterior parietal cortex (Wilbur et al., 2014), retrosplenial cortex (Alexander et al., 2020), postrhinal cortex (LaChance et al., 2019), and the striatum (Hinman et al., 2019). While the introduction and discussion of this manuscript mainly focus on literature related to social interaction, no clear control experiments were done that would separate this type of egocentric social vector tuning from egocentric vector tuning to any other goal, object, or reference point. If experiments demonstrated this tuning is stable and is indeed social, it would be a significant advance for the field.

Other comments:

6. Line 40 of the Introduction states: "Object-vector cells in the hippocampus encode the position of an object relative to the animal, irrespective of the position of the object within the environment⁶⁻⁹. One synapse upstream, the medial entorhinal cortex has object-vector cells with similar characteristics¹⁰" The referencing here is inaccurate and confusing. Of the four papers cited claiming that the hippocampus contains object-vector cells, only Nagelhus et al claims that. Others merely report different types of changes due to introducing objects. In addition, Deshmukh & Knierim (2013) were the first to report vectorial coding to objects in CA1 cells and should be cited.

7. A more informative decoding would be to report the number of cells that you need to achieve an error significantly below chance. In addition, it would be nice to decode the angle or distance separately for the vectorial responses. As it is, a decoding error could result from purely an error in either angle or distance.

8. It is incorrect and unacceptable to describe results as "modest" (Figure 3) simply because the P values are not as low as other results in the paper.

9. It is unclear how many cells were recorded per session and whether the cells were the same across days. It is mentioned that a method was used to track cells across sessions, but where is this information used? If comparing cells across sessions is important for the claims, the imaging footprints of example cells should be shown to demonstrate confidence in declaring the cells were the same.

10. The behavioral videos of mouse interactions, the DeepLabCut analysis, and examples of manual annotations of specific behaviors should be presented. That analysis provides a rich context of the social behavior, but as presented here, it is all dismissed by the authors.

11. In Figure 4, the number of egoSVCs appears to be negatively correlated with environment size, suggesting that small environments are biased toward finding more vectorial responses. This should be noted and addressed in some way.

12. It is not obvious that the pursuit task forces the mouse to do something complex and social, rather than just running in circles and/or chasing a reward. A nice control would be chasing an object (e.g., a fishing task, or a laser-chasing task). Without this control, one could argue that any observed vectorial coding could be attributed to the animal receiving/tracking a reward instead of responding to the conspecific.

13. In Figure 5, the sampling gets much worse due to the repetitive behavior, especially for trained animals. It should be shown that the sharpening of tuning is not a result of biased sampling. If this cannot be shown, the conclusion is invalid. The same problem applies to Figure 6.

14. The title is too general and should specify that the results are limited to area CA1 of the hippocampus, and that only female mice were tested.

15. It would be very helpful to label the coordinates of the rate maps for each reference frame. For example, one could use cardinal directions for allocentric maps and

“front, back, left, right” for egocentric maps. It would also be useful to show more clearly where the head of the animal is centered on the egocentric plots.

16. In some figures, the authors use an asterisk to indicate which cells had spatial information values that exceeded chance levels. This helpful label should be used everywhere.

17. Occupancy criteria for rate maps should be clearly defined and more conservative. Criteria should include multiple passes through each spatial, distance, and/or directional bin.

18. The methods should clearly specify how the data was split for cross-validation. It is recommended to use several small chunks of data at the relevant behavioral timescale instead of taking every 5th time bin, for example.

19. Chance levels are not defined in the results section. While there may be a field standard (e.g., exceeding the 95th percentile of a shuffled distribution), it should be clearly stated within the results and methods section.

In this study, the authors have undertaken a comprehensive exploration of hippocampal representations of a conspecific performing a spatial task. The research has identified two distinct cellular responses that encode the positions of others relative to the observer (referred to as "social-vector cells") in both allocentric and egocentric coordinates, shedding light on the neural mechanisms underlying social interactions. Furthermore, the authors have made an intriguing observation that learning a pursuit task enhances the tuning of social-vector cells, although it does not affect their numerical count. The manuscript is well-structured and written, reflecting a meticulous and thoughtful research effort. Nevertheless, several critical questions and points merit attention and clarification before its acceptance.

Major comments:

1. In this study, egocentric social-vector cells are characterized by their reliance on self-head orientation as a defining factor. However, we are intrigued by the possibility of adopting body orientation instead. What would be the implications if body orientation were chosen as the reference point?
2. In this study, the authors employed deconvoluted calcium activity in conjunction followed by binarization for individual cells. It's noteworthy

that some research groups solely utilize deconvolved calcium activity without employing binarization. Given the inherent uncertainty associated with calcium activity due to noise and variations in extraction algorithms, it is imperative to acknowledge the potential introduction of artifacts stemming from binarization. It is recommended that the manuscript addresses the specific threshold value employed for the binarization process to provide clarity and transparency regarding this critical methodological aspect.

3. Additionally, we are curious to understand whether the study explored different outcomes by exclusively utilizing deconvolved calcium activity without the incorporation of binarization.

4. In Figure 1j, there is no statistically significant distinction in the cross-validated decoding error between social-place cells and the shuffled data. Please explain that.

5. Again in Supplementary Figure 4, no statistically significant distinction in the cross-validated decoding error between social-place cells and the shuffled data is observed.

6. Kindly explain the criteria used to pinpoint the specific moments for calculating allocentric and egocentric social-vector cells based on head orientation. I am particularly interested in understanding the chosen temporal instances during which the mouse is actively encoding spatial information related to conspecifics, while excluding intervals associated with alternate behaviors such as foraging and sniffing.

7. In line 169, the authors mentioned that “coding of conspecifics is not directly related to the amount of interaction within the environment”. To enhance the comprehension of this statement, it would be beneficial to provide a more detailed explanation, as the manuscript does not currently offer conclusive evidence.

Minor comments:

1. The manuscript lacks specific information regarding the GRIN lenses employed in the study. By checking Supplementary Figure 1, it is likely that there are variations in lens diameters, with some appearing to be approximately 1mm in diameter, while others appear to be closer to 2mm. I kindly request clarification on this aspect.

2. Could you please provide clarification regarding the choice of miniscopes used for imaging: UCLA Miniscope V3 or V4?

3. In some cases, spread depression can indeed affect calcium imaging activity in the CA1 region when using GRIN lenses and Miniscope systems. I am interested in whether this potential issue was encountered or addressed during the course of this study.

4. To enhance clarity, please provide clarification on whether the head orientation of the mice was determined or extracted using the DeepLabCut software in this study.

Multiplexed representation of others in the hippocampal CA1 subfield of mice

Point-by-point response to reviewers.

In black font: comments by reviewers.

In blue font: our replies.

In red font: major modifications introduced to the manuscript to address specific comments.

REVIEWER COMMENTS

Reviewer #1 (Remarks to the Author):

The authors of this paper recoded social-spatial representations in CA1 area of the mouse hippocampus, in various behavioral tasks, and using calcium imaging. They report single-cell representations of social-place cells, allocentric social-vector cells, and egocentric social vector cells.

This is an important study which brings additional supporting evidence for new perspective on the mammalian hippocampal circuits – their involvement in social behavior.

The paper is well written, and the analysis is concise, yet I am not convinced by some of the results and the interpretations made by the authors in this paper.

Comments

1. Using the classical single cells tuning maps of place cells to untangle spatial representations of self, form spatial representation of others in CA1 is challenging. It is possible that correlated movement patterns between self and conspecifics may cause artifactual social-spatial representation. For example, a classical place cell representation could be misinterpreted to a social-spatial representation in an extreme case when the movement of the recorded mouse is correlated with the movement of conspecifics (for example when one mouse is chasing another mouse). The manuscript completely ignores this possibility, and no evidence is presented to rule this out. I suspect that the proportion of the different cell categories would drastically change after taking this into account, and this of course will directly impact the results and conclusions from the conjunctive coding analysis, the population analysis, etc. The authors need to add compelling evidence that their social place cells and social vector cells are not contaminated by self-spatial responses and cannot be explained by stereotypically correlated movements patterns.

We agree with the reviewer that this alternative explanation of our findings needs to be ruled out. To do so, we present two independent arguments. First, note from Figure 1 that more than half of cells classified as egoSVC are not classified as selfPC. One would not expect to see this if cells were responding to the position of the observer and only by chance this coincided with a relative position of the other mouse in the coordinates of the observer's head direction. Furthermore, note from Figure S3h that the overlap between both groups of cells is not larger than expected by

chance (roughly half of the whole population and also roughly half of egoSVCs have place cell-like spatial modulation). Again, this overlap should be much higher than expected by chance if self-place was the cause of egoSVC activity.

As an independent argument, in a new set of analyses we address the possibility that egoSVC responses are explained by correlated movements. To do so, we identified the moments in which the relative velocity between mice (one velocity vector minus the other) had a low vector length and left them out of the analysis. This equates to removing from the analysis moments in which both animals execute similar movements. We repeated the classification of cells of Figure 1 considering only relative speeds above 2,5 cm/s (which removed $12 \pm 2\%$ of data) and above 5 cm/s (which removed $26 \pm 3\%$). In both cases we obtained percentages of cells very similar to the original ones, ruling out the idea that correlated movements are responsible for a given cell type.

These ideas are now included in Figure S4a and discussed in the Results section:

“In particular, chance-level overlap between the selfPC and egoSVC subpopulations made it unlikely that the firing of the latter could be explained by the position of the observer relative to the arena. Furthermore, percentages of cell types did not vary substantially after removing episodes during which the movement of both animals was correlated (Supplementary Fig. 4a).”

2. I would also like to see some quantification of the movement patterns and coverage of the behavior in the social-vector cells. I suspect that biased coverage of head direction angles might also introduce artifactual social-vector representations.

We agree that this is an important issue, also raised by other reviewers. We have added several analyses and plots aimed on one hand to quantify coverage and on the other hand to demonstrate that our results are not favored by poor coverage.

First, Figure 1h-i now shows coverage as a function of relative distance between mice for alloSVC and egoSVC perspectives. It is clear from these plots that coverage decreases close to the edge of the effective arena. This is because the condition to achieve the maximum possible distance between mice is that they stand exactly in opposite poles of the arena, which rarely happens. Coverage at smaller distances is not subject to similar constraints. Note that the examples of best egoSVC coverage now shown in Figure S4e all have a coverage around 85 % despite no apparent ‘holes’, which is explained by the fact that what is missing is the outer ring of the effective arena. Importantly, the new Figure 1h-i shows that no social vector fields are identified in this region of poor coverage, which suggests that coverage is, if anything, biasing results toward an underestimation of the number of social vector fields and cells.

Second, Figure S4a now shows a re-classification of cells under two restrictive coverage conditions. On one hand, for all sessions that had a coverage above 70 % in the coordinate system where maps are being analyzed. On the other hand, for all sessions that had a coverage above 70 % in all coordinate systems simultaneously (8 out of 18; 4 examples shown in Fig. S4e).

In both cases we found that the classification of cell types was very similar to the one obtained including all sessions, which suggests that our results are not being biased by 'holes' in coverage.

Third, regarding a potential role played by poor head direction coverage, we re-classified cells using only sessions where head-direction mean vector length was below 0.1 (5 out of 18 sessions; head direction, egoSVC direction and egoSVC fields now shown in Fig. S4b). Again, we found that the classification was very similar to the one obtained from all sessions (Fig. S4a).

Together, these new analyses suggest that social vector coverage and head direction coverage are not biasing our results, beyond potentially causing that we do not identify SVC fields close to the edge of the effective arena.

We have added to the caption of Figure 1h-i:

“Note that around 70 cm, the maximum possible distance between mice, field density and coverage decay because conditions for animals to be at exactly 70 cm from each other are rarely met.”

3. The authors conclude and generalize in line 106 – that their results point to a relative rather than absolute position of social-spatial coding in CA1. This extrapolation to all mammalian CA1 hippocampus is not supported by their results. Previous studies on social place cells recorded from stationary observer while observing a moving conspecific, and from a different species (rats and bats). The most immediate and obvious possible explanation for the difference between their results and previous studies is a simple species-specific behavioral difference. This should be made explicitly in the text.

We agree that in the original wording we were extrapolating without evidence. We now restrict our conclusions specifically to mice. This said, we want to stress that, as stated in the discussion, we find that our results are compatible with these groundbreaking studies in other species. While a difference between species is possible, it is not necessary to harmonize results. This is precisely because previous experiments were done on mostly stationary observers.

We modified line 106 but also added in the first paragraph of the discussion:

“, although potential differences between species in this aspect of CA1 coding require further assessment.”

4. In relation to the social vector cells, I am missing quantification of the distribution of the distance-to-conspecific across social vector cells (ego and allo)- is it confined to small distance range or evenly distributed across all recorded distances?

We agree with the reviewer on the obscurity around this issue and have modified Figure 1h-i and its caption to address it. The relative distance between animals covers homogeneously most of the range of possible distances (0 to 70 cm), as the new Figure 1h-i shows. It is now also highlighted that the arena is 70 cm in diameter. The coverage decays close to the edge for both alloSVC and egoSVC perspectives. These extreme values of relative distance are only covered when both animals stand at perfectly opposite poles of the arena, making them very improbable, while less extreme values of relative distance do not have this kind of constraint.

Regarding the relative distance to the center of egoSVC and alloSVC fields, we also find a homogenous distribution that decays close to the boundary of the effective arena (Fig. 1h-i). This is likely a direct consequence of low coverage in this region, as explained in the previous paragraph.

The third paragraph of results now reads:

“The density of peaks of social-vector fields was homogeneous across most of the effective arena and only decayed close to the edges, where coverage also decreased (Figs. 1h-i).”

5. In line 91 and in Figs. 1e, supp. 2a – it is not clearly explained what do the authors mean by “effective arena”, and why the “effective arena” is bigger for social vector cells?

We think that the concept of effective arena is important and agree that it was insufficiently developed. We have introduced a new supplementary figure to explain it. Imagine an arena that is 1 m long along the S-N direction. When the observer stands at the S tip and the other mouse stands at the N tip, this translates in alloSVC terms into 1 m in the N direction (or its equivalent in egoSVC coordinates). In the opposite situation, when the observer is in the N tip and the other mouse in the S tip this translates into 1 m in the S direction in alloSVC coordinates. Effectively, SVC coordinates span from 1 m S to 1m N, an overall 2 m range, while coordinates in the physical arena span over a range of 1 m. Since the same happens in the W-E direction, the area of the effective arena is 4 times as large as the physical arena. We have now made this clear in the new Figure S3a and added the following explanation in the Results section:

“With these definitions, the space representing all possible relative positions (the ‘effective arena’) exhibited dimensions twice as large as those of the physical arena (4 times in terms of area), so spatial bins were also twice as large (Supplementary Fig. 3a).”

6. In the pursuit tasks – the egoSVC is collapsed to a single angle on the track- if I understood correctly, there is an assumption that the head direction and the heading direction are similar in this task. If this is correct, then I would like to see some analysis of the head direction during pursuit -so as to see if this assumption holds true.

The reviewer is correct in that this assumption is made in Figure 6. We included this approximation because it allows for a better visualization and intuitive understanding of maps, but note that Figure 5 reaches similar conclusions without any assumptions. We are including Figure S11a, which shows that the assumption of parallel track and head directions is correct most of the time.

7. Fig. 6e is interpreted as sharpening the tuning – but to me it seems that instead of sharpening – the DC activity is lowered. If one would scale the tuning curve to min/max then it seems that the tuning is not sharpened but instead, there is a decrease in baseline of the tuning curve – which would mean an increase in SNR instead of sharpening of the tuning curve.

We agree with the reviewer's observation on plot shapes, but want to note that Figure 6d-e shows correlation, not neural activity, as is now highlighted in the new version of the Results section. We chose this measure because the decay in the autocorrelation of a map is typically used as a proxy of the field size that applies even when multiple fields are present. Strictly speaking it describes how far one needs to move to have a strong change in population code, and the minimum value speaks of the overall strength of this change. We have re-worded the text to provide a better interpretation of the results.

This said, we find interesting the perspective in terms of signal and noise and added new analyses in this direction. We defined the signal of every map as the region defined by the decay to half around the maximum event rate bin, and the noise as the rest of the map. We added two panels in Figure 6c showing that a) the signal region is smaller for trained animals and b) the ratio of activity corresponding to signal vs. noise parts is higher. We believe that these analyses address the concerns of the reviewer.

We added to the Results section:

“We also defined for each map a signal region, including the peak event rate bin and the collection of bins around it defined by half-decay in event rate, and the complementary noise region. We observed that the signal region was smaller for trained animals, although it concentrated a higher fraction of the overall cell activity (Fig. 6c). These observations suggest that as animals learn the task oSVCs become sharper and more selective.”

Reviewer #2 (Remarks to the Author):

The authors set out to understand the nature of the hippocampal representation of conspecific animals. It has been shown several times that hippocampal neurons respond to the position of not only the animal's current location but also the location of a conspecific. It is unclear, however, whether the representation of others is in a world-centered or self-centered reference frame because previous experiments recorded neural activity from a passive observer. Here, they allow

female mice to freely interact or engage in a pursuit task. In addition to classical place cells (selfPC) and place cells representing the location of others (socialPC) they report two new classes of hippocampal CA1 neurons with vector-based responses to the conspecific in either allocentric or egocentric reference frames. In addition, they state that various factors, including mouse identity and familiarity, environmental context, and task modulate these responses.

This study asks an important and timely question of how animals track the location of others during social interactions. Recent technological advances in lightweight recording devices and quantitative behavioral tracking have made it possible to address such questions. The variety of experimental conditions and the ability to record large numbers of CA1 neurons across sessions using calcium imaging have yielded an impressive and valuable dataset. The figures are nicely presented with many example responses, and individual data points are used to show outcome variability clearly.

Major comments:

1. To claim a new functional cell type exists requires substantial evidence. Such evidence is lacking here. Cells were categorized into four non-mutually exclusive functional classes based on two criteria: spatial information content relative to a shuffled distribution and within-session map stability with a Pearson correlation > 0.3 . These are liberal criteria relative to what is commonly used in hippocampal literature. More rigorous criteria additionally include a minimum firing rate, a detectable firing field, and most importantly, stability between separate recording sessions. Between-session stability (usually assessed by Pearson correlation > 0.5) is critical to eliminate false positives due to biased behavioral sampling or confounding cues present in just one session. Without showing that these vectorial responses produce stable firing fields across different recording sessions, one cannot conclude these functional classes exist. Because the entire paper hinges on the existence of these new cell classes, this point must be addressed.

We are addressing this concern from multiple complementing perspectives. First, we added Figure S4a, which shows a re-classification of cells as in Figure 1 but using stricter criteria: mean event rate higher than 0.1 Hz and a detectable firing field larger than the equivalent to 5 bins x 5 bins. We found a distribution of cell types that is very close to the original one. Regarding the 0.5 correlation cutoff, our re-classification also shown in Figure S4a suggests it is too restrictive for our specific data (binarized calcium signals obtained with miniscopes in mice running for 10 min). Percentages of cells types are reduced in a rather proportional way, suggesting that no cell type is particularly favored by the choice of correlation threshold. EgoSVCs still represent 11 % of the population, but classical place cells are below 20 %. While we feel that our original choice of threshold harmonizes better our results with the literature, the new analysis shows that still a substantial number of cells are classified as egoSVC with a 0.5 threshold, ruling out that social vector cells are an artifact caused by the threshold choice.

Second, to address cross-session stability, we performed experiments on 2 new pairs of mice. We compared cross-session stability of all cells according to their classification in the first session, as now shown in Figure S4c. We observed that all cell types had a stability distribution higher

than expected by chance (shuffling) with the exception of socialPC. This result goes in the same direction as some of our previous results, suggesting that socialPC coding is poor in the CA1 subfield of mice. We compared the cross-session stability distributions for different cell types and found that they were quite similar to each other with the exception of socialPC. Within the other 3 cell types we only found a significant difference between alloSVC (which had the lowest cross-session stability) and selfPC (which had the highest). Importantly, the cross-session stability of egoSVC and selfPC was not significantly different. While we cannot assess the cross-session stability for the bulk of our data because we did not perform multiple sessions with the same partner, these new datasets and analyses suggest that the cross-session stability of egoSVCs is similar to that of other well-established cell types.

2. Related to the previous comment, sampling space in both allocentric and egocentric coordinates is difficult to achieve in short 10-minute sessions. This problem is exacerbated by the 5-minute within-session comparisons used to claim tuning stability. The sampling is especially problematic because of the potential interaction between the two reference frames. In other words, an animal may visit all places in the environment and all vectors relative to the conspecific but will rarely visit all combinations of the two (e.g., north of the conspecific while being in the south end of the arena). This is visible in some examples in Supplemental Figure 2. This biased sampling will necessarily confound the interpretation of maps using one reference frame while ignoring the other reference frames. This could be addressed by plotting the coverage of the space in several reference frames against each other. The authors should require that coverage and occupancy in the other three reference frames also meet some acceptable threshold before classifying the cell. Additionally, the authors must show maps in all reference frames for each cell that meets the criteria for one or more of the defined cell types.

We agree with the reviewer that coverage is important and was not sufficiently addressed. We present here an image plot of coverage in different coordinate reference systems for all sessions in Figure 1, which we have not included in the new version of the manuscript because we feel it is too specific (but would be willing to reconsider).

The figure shows a high correlation in coverage between pairs of coordinates whenever neither of them is socialPC ($R > 0.7$). In particular, egoSVC and selfPC have an R of 0.78 and most sessions are above 70% of coverage in both coordinate systems. Note also that egoSVC and alloSVC never reach values close to 100%. This is because the edges of the effective arena have very poor coverage (see new version of Fig. 1h-i). They can only be reached in specific situations when both animals are in perfect opposite poles of the arena, and this condition is rarely met (note the examples with best coverage in Figure S4e, which only go up to around 85 % despite no observable ‘holes’ in the coverage). This means that for SVCs our observations at the edge of the effective arena might be biased by low coverage. Since the detection of fields falls to zero in this region (new Fig. 1h-i) this might mean that we are missing some SVC fields close to the edge, implying that poor coverage does not favor our results.

To investigate a more general potential bias in our results due to coverage, we selected only sessions that had a coverage above 70 % in a) a given coordinate system or b) all coordinate systems simultaneously. The procedure b) led us to select the best 8 out of 18 sessions (4 examples shown in Figure S4e). We found that the percentages of cells (see new version of Fig. S4a) of each type were in both cases very similar to the original ones, which suggests that coverage is not biasing our results in a general way.

3. Decoding of position is used in this study to justify that information about spatial location is present in different populations of cells. The decoding error is higher than expected in Figure 1J for traditional place cells (selfPC), where approximately half of the sessions are shown in this plot to have a decoding error comparable to the Shuffle group (ca. 20 cm). Why is the decoder so inaccurate using standard place cells? It would be insightful for the authors to investigate why this error rate is so high. Additionally, the decoding error is substantially worse for all other cell classes relative to their respective shuffled groups, and significant differences between the true distribution and shuffled distribution are likely due to such a large sample size (see next comment).

CA1 place cell decoding error in absolute terms depends on many factors, such as the species, the recording technique and the geometry and size of the environment. Most studies use one-dimensional tracks, where decoding is more accurate than in the open field for geometrical reasons. Errors also increase with the size of the arena and depend on the number of recorded neurons. Among reports of decoding in two-dimensional arenas, a study by Frey and colleagues (2021) in rats running in a 175x125 cm rectangular arena reports a median error between 25 cm and 80 cm when using between 30 and 1 tetrodes, respectively. Another work by Stefanini and colleagues (2020) is more similar to ours in that it studies the CA1 subfield of mice using Inscopix endoscopes. This work reports median errors between 9 and 15 cm in a 50x28 cm rectangular arena. Another work by Etter and colleagues (2020) optimizes the hyperparameters of a Bayesian decoder for miniscope CA1 recordings in mice running in a 45 cm side square. They found that the best combination of parameters yields a mean error of around 13 cm. We believe that our results showing a mean error of about 15 cm in a 70 cm square arena, with some sessions showing an error below 10 cm, are comparable to the cited studies.

To further describe decoding errors in our recordings, we asked if it decreased with the number of neurons. We found that this was indeed the case for all cell types with the exception of socialPCs, as is now presented in Figure S5c. This shared tendency suggests that smaller effect sizes for egoSVCs and alloSVCs relative to selfPCs are related to the fact that recorded cell subpopulations are smaller. It should also be noted that the effective arena for egoSVCs and alloSVCs is twice as large as the physical arena (4 times the area; now explicitly discussed in the Results section and shown in Fig. S3a) and errors are expected to be around twice as large as well.

References:

Frey, M., Tanni, S., Perrodin, C., O'Leary, A., Nau, M., Kelly, J., Banino, A., Bendor, D., Lefort, J., Doeller, C.F. and Barry, C., 2021. Interpreting wide-band neural activity using convolutional neural networks. *Elife*, 10, p.e66551.

Stefanini, F., Kushnir, L., Jimenez, J.C., Jennings, J.H., Woods, N.I., Stuber, G.D., Kheirbek, M.A., Hen, R. and Fusi, S., 2020. A distributed neural code in the dentate gyrus and in CA1. *Neuron*, 107(4), pp.703-716.

Etter, G., Manseau, F. and Williams, S., A probabilistic framework for decoding behavior from in vivo calcium imaging data. *Front Neural Circuits*. 2020; 14: 19.

4. The interpretation and reporting of statistical results, specifically regarding P-values, is flawed and could be improved by accounting for additional details like effect size. For example, Figure 1J uses P values from a Mann-Whitney test between two samples. These P values are extremely small, leading the authors to conclude there is a significant difference between the data and the shuffled groups, and therefore, these functional classes contain meaningful levels of information. This interpretation is flawed because it does not examine the effect size. Having small P values is completely meaningless on its own because if one collects enough samples the P value gets

smaller by definition. This problem persists throughout the paper and undermines the majority of claims. Effect size must be reported everywhere, along with other statistical details, including sample size, degrees of freedom, and test statistics.

We agree in part with the reviewer, but want to note that it is not the general case that P-values are small by definition if one collects enough samples. This only happens if the distributions from which the two groups of samples are drawn are intrinsically different, which is what the statistical test intends to demonstrate. We have modified the manuscript to include effect size, degrees of freedom and test statistics as required.

For example, in Figure 1j we added Cliff's Delta values using MATLAB function "*meanEffectSize*":

"... j, Cross-validated decoding error in each reference frame compared with shuffling (one dot per session; mean \pm s.e.m., Mann Whitney test p-value indicated. From left to right, session number: 321, 182, 295, 338; Mann-Whitney U: 1779716, 1137944, 2432712, 3255215; Cliff's Delta: -0.65, -0.31, -0.44, -0.43)."

5. The authors fail to cite and discuss several relevant studies and thus do not place their work in appropriate context. Egocentric coding in the hippocampus has been reported in bats (Sarel 2017), mice (Jercog et al., 2019), and rats (Ormond et al., 2022). Egocentric coding similar to the type discussed here has additionally been found in several brain areas important for spatial navigation, including posterior parietal cortex (Wilbur et al., 2014), retrosplenial cortex (Alexander et al., 2020), postrhinal cortex (LaChance et al., 2019), and the striatum (Hinman et al., 2019). While the introduction and discussion of this manuscript mainly focus on literature related to social interaction, no clear control experiments were done that would separate this type of egocentric social vector tuning from egocentric vector tuning to any other goal, object, or reference point. If experiments demonstrated this tuning is stable and is indeed social, it would be a significant advance for the field.

We have now cited this literature in the Introduction. Regarding the coding of objects or goals vs conspecifics, we cannot fully discard that some common mechanisms are being used, and an exhaustive comparison of coding strategies for objects, goals and conspecifics could be interesting for a follow-up project. The main novelty in our manuscript is the prevalence of vector coding for representing others in the CA1 subfield of mice, while the only alternative strategy that had been explored, socialPC, is marginal. This said, we have found some elements that seem to differentiate egocentric coding of objects and of others in the CA1 subfield of mice. In new data obtained from two mice, we compared cross-session stability of both types of responses. In one case, mice foraged in the presence of the same conspecific during two consecutive sessions. In the other case, mice foraged in the presence of an object that on the second session was placed in a new location. We found that egocentric object vector responses had a cross session stability that was significantly lower than that of egoSVCs. These new data point to differences in the strategy for coding objects and others, although a full characterization would require many more specifically designed experiments. The focus of our manuscript is on comparing strategies in

terms of coordinates systems that mice use for social coding, and we feel that the question of whether or not and to what degree similar strategies are used for coding objects in CA1, while in many aspects still an open one, is secondary in this context. These results are now shown in Figure S4d.

Other comments:

6. Line 40 of the Introduction states: "Object-vector cells in the hippocampus encode the position of an object relative to the animal, irrespective of the position of the object within the environment⁶⁻⁹. One synapse upstream, the medial entorhinal cortex has object-vector cells with similar characteristics¹⁰" The referencing here is inaccurate and confusing. Of the four papers cited claiming that the hippocampus contains object-vector cells, only Nagelhus et al claims that. Others merely report different types of changes due to introducing objects. In addition, Deshmukh & Knierim (2013) were the first to report vectorial coding to objects in CA1 cells and should be cited.

We agree and have modified the Introduction.

7. A more informative decoding would be to report the number of cells that you need to achieve an error significantly below chance. In addition, it would be nice to decode the angle or distance separately for the vectorial responses. As it is, a decoding error could result from purely an error in either angle or distance.

We added this idea to new Figure S5c. Unfortunately, for most cell types a single cell is enough to differentiate decoding from what would be obtained from shuffled distributions. However, this new panel interestingly shows that for most cell types the error decreases with the number of neurons used for decoding. The only exception is socialPCs which, as many other results in our manuscript suggest, are either very few or present a code that is less robust in comparison.

Regarding the decoding of angle and distance, we have added new plots (Fig. S5d) that show that they can both be decoded separately.

8. It is incorrect and unacceptable to describe results as "modest" (Figure 3) simply because the P values are not as low as other results in the paper.

Our intention was to contrast the huge differences in behavioral patterns (> 3 fold higher interaction time) with what we had expected that could have been higher differences in spatial information. However, this was not clear at all from the title of Figure 3, so we have changed it to something more neutral: "EgoSVC representation of familiar and unfamiliar conspecifics". We have also changed the wording in the description of results.

9. It is unclear how many cells were recorded per session and whether the cells were the same across days. It is mentioned that a method was used to track cells across sessions, but where is this information used? If comparing cells across sessions is important for the claims, the imaging footprints of example cells should be shown to demonstrate confidence in declaring the cells were the same.

We agree that clarity lacked around this issue. Tracking of cells across sessions was used in Figures 3 and 4, as is now described in the Methods section. We also include examples of tracked cells in the supplementary figures associated to Figures 3 and 4 (Figs. S7 and S8). And we have included cell numbers in all figure captions.

10. The behavioral videos of mouse interactions, the DeepLabCut analysis, and examples of manual annotations of specific behaviors should be presented. That analysis provides a rich context of the social behavior, but as presented here, it is all dismissed by the authors.

We have now added two example videos included as Supplementary Material showing simultaneously DeepLabCut tracking and annotations of behavior.

11. In Figure 4, the number of egoSVCs appears to be negatively correlated with environment size, suggesting that small environments are biased toward finding more vectorial responses. This should be noted and addressed in some way.

We agree that this relationship is apparent in Figure 4. However, as we now show in a quantitative assessment, the proportion of cells classified as alloSVC and egoSVC does not change significantly across environments of different size or geometry, as is now shown in Figure S8h.

12. It is not obvious that the pursuit task forces the mouse to do something complex and social, rather than just running in circles and/or chasing a reward. A nice control would be chasing an object (e.g., a fishing task, or a laser-chasing task). Without this control, one could argue that any observed vectorial coding could be attributed to the animal receiving/tracking a reward instead of responding to the conspecific.

We partly agree with the reviewer that chasing could be a general mechanism, although note that the reward is not being chased in this case. It rather comes as a consequence of chasing a conspecific and is delivered on a different region of the maze, towards the back of the pursuit trajectory (Fig. S9). This concern has some elements in common with point 5 raised by the reviewer. At this point, we cannot claim that the same cells or mechanisms that are used to represent conspecifics could not be also used to chase inanimate objects. Note, however, that in our response to point 5 we present some hints of a difference between the coding of objects and conspecifics. This said, we feel that the question of the reference coordinate systems in which

mice code for conspecifics is still novel and interesting, even if a future study finds that similar mechanisms are used for chasing objects that move. In this context, Figures 5 and 6 show that social vector coding, which emerges spontaneously to represent conspecifics (as shown by introducing unfamiliar ones) can be refined when a reward associated to pursuit is introduced. Note that it is not the number of cells that change, so that it is unlikely that new generic 'pursuit' cells are added to the representation when the animal learns the task, but instead the egoSVC tuning becomes sharper.

13. In Figure 5, the sampling gets much worse due to the repetitive behavior, especially for trained animals. It should be shown that the sharpening of tuning is not a result of biased sampling. If this cannot be shown, the conclusion is invalid. The same problem applies to Figure 6.

We agree with the reviewer and have repeated analyses in Figure 5 using only the 50% of spatial bins with highest coverage for the average of trained animals (roughly a figure '8', top and bottom loops representing counter clock- and clockwise trajectories, respectively). The results show that this manipulation not only does not suppress but actually slightly enhances the differences we originally found, ruling out that they are caused by stereotypic movement patterns in trained animals. The same applies to information differences in Figure 6. These results, are presented in Figure S11b-d.

14. The title is too general and should specify that the results are limited to area CA1 of the hippocampus, and that only female mice were tested.

We changed the title to: "Multiplexed representation of others in the hippocampal CA1 subfield of mice". We assessed the possibility of using the title "Multiplexed representation of others in the hippocampal CA1 subfield of female mice". In our opinion, given the history of publications in the field that only use male animals and only mention it in the Methods section, this title would give the wrong impression that our finding contrasts what we found in female mice to what happens in males. This kind of representation has not been assessed in males and it could be the case that it is very similar to the one we find in females. Instead, and in agreement with journal policies, we are now including in the abstract the fact that only female mice are used. This said, we would be willing to further consider the inclusion of gender in the title and discuss pros and cons with the reviewer and/or the editor if the reviewer feels it is necessary.

15. It would be very helpful to label the coordinates of the rate maps for each reference frame. For example, one could use cardinal directions for allocentric maps and "front, back, left, right" for egocentric maps. It would also be useful to show more clearly where the head of the animal is centered on the egocentric plots.

We have taken this approach in Figure S3a, which now describes the effective arena for alloSVCs and egoSVCs. Although it would be nice to have it in all figures, we do not see how we could introduce it without taking up too much space.

16. In some figures, the authors use an asterisk to indicate which cells had spatial information values that exceeded chance levels. This helpful label should be used everywhere.

We have included asterisks in all figures.

17. Occupancy criteria for rate maps should be clearly defined and more conservative. Criteria should include multiple passes through each spatial, distance, and/or directional bin.

This concern is similar to major point 2 raised by the reviewer. To summarize the answer, Figure S4a shows that the classification of cells with stricter coverage criteria is similar to the original one, suggesting that the results are not biased by uneven coverage.

18. The methods should clearly specify how the data was split for cross-validation. It is recommended to use several small chunks of data at the relevant behavioral timescale instead of taking every 5th time bin, for example.

We agree with the reviewer and the suggested approach is the one we actually took, although it was not clear in the original Methods section. We have now made it clear:

“To do this, rather than randomly selecting data, sessions were divided into 5 consecutive segments, so that segment 1 corresponded to the initial part of the session and segment 5 to the final part.”

19. Chance levels are not defined in the results section. While there may be a field standard (e.g., exceeding the 95th percentile of a shuffled distribution), it should be clearly stated within the results and methods section.

We agree and have added to the Results section when first introducing the classification:

“allowing for cells to be included in multiple categories simultaneously (Fig. 1e and Supplementary Fig. 3b; cutoff value for classification: 95th percentile of the shuffled distribution for information).”

In this study, the authors have undertaken a comprehensive exploration of hippocampal representations of a conspecific performing a spatial task. The research has identified two distinct cellular responses that encode the positions of others relative to the observer (referred to as "social- vector cells") in both allocentric and egocentric coordinates, shedding light on the neural mechanisms underlying social interactions. Furthermore, the authors have made an intriguing observation that learning a pursuit task enhances the tuning of social-vector cells, although it does not affect their numerical count. The manuscript is well- structured and written, reflecting a meticulous and thoughtful research effort. Nevertheless, several critical questions and points merit attention and clarification before its acceptance.

Major comments:

1. In this study, egocentric social-vector cells are characterized by their reliance on self-head orientation as a defining factor. However, we are intrigued by the possibility of adopting body orientation instead. What would be the implications if body orientation were chosen as the reference point?

We agree with the reviewer that this is an interesting point. Head direction rather than body direction is traditionally used because in rats these two variables may differ substantially and head direction explains better neural firing in the hippocampus and surrounding areas (possibly because most senses rotate with the head). In mice, we find that head direction and body direction do not differ so much. We trained DeepLabCut to estimate both variables in a session. Head direction, as in the manuscript, was obtained from LEDs rotating with the head. For body direction we used as markers the point where the tail meets the body and the point between the ears. It should be note that this measure fails to capture the potential curvature of the body. We found that the difference between head and body angle is tightly centered around 0, as shown in the following plot.

In agreement with this observation, we found that the information content of egoSVCs was not significantly different when using body instead of head direction to calculate it (Man-Whitney test, $p: 0.75$). We decided not to include these results in the manuscript because we feel that the coding of the body by the hippocampus deserves a more in-depth and nuanced investigation and our results are only confirming the traditional approach. However, we would be willing to reconsider if the reviewer feels it is important.

2. In this study, the authors employed deconvoluted calcium activity in conjunction followed by binarization for individual cells. It's noteworthy that some research groups solely utilize deconvoluted calcium activity without employing binarization. Given the inherent uncertainty associated with calcium activity due to noise and variations in extraction algorithms, it is imperative to acknowledge the potential introduction of artifacts stemming from binarization. It is recommended that the manuscript addresses the specific threshold value employed for the binarization process to provide clarity and transparency regarding this critical methodological aspect.

We agree with the reviewer and have now clarified that we use 3 standard deviations as a threshold for binarization. Although we understand that this is a rather conservative choice, we now show in Figure S4a that the results of cell classification are very similar to the ones obtained using a threshold of 0 standard deviations (signal higher or lower than the mean) or even directly using the deconvoluted signals without binarization. This similarity suggests that our results are robust with regard to the choice of parameters to process calcium signals. In addition to the description in the Methods section, the first paragraph of results now reads:

“To study social spatial representations in the hippocampus, we recorded calcium events (binarized with a cutoff of 3 standard deviations) from GCaMP6s-expressing ...”

3. Additionally, we are curious to understand whether the study explored different outcomes by exclusively utilizing deconvoluted calcium activity without the incorporation of binarization.

New Figure S4a now shows that cell classification is similar to the original one if no binarization of the deconvoluted signals is used, which suggest that these cell types can be investigated with any of these approaches. The figure also shows examples of maps obtained with the different methodologies (Fig. S4f). Although the maps have differences, classification by comparison with shuffled statistics seems to be compensating for them.

4. In Figure 1j, there is no statistically significant distinction in the cross- validated decoding error between social-place cells and the shuffled data.

Please explain that.

While the size effect is rather small, we actually do find a significant improvement in decoding for this group, relative to what is obtained with shuffled data ($p: 3.5 \cdot 10^{-13}$). While at the single cell level socialPCs are not a significantly sized subpopulation, at the population level their activity is enough to produce this very small improvement in decoding. Our explanation for the difference between socialPCs and vector social representations is that the later are favored to represent conspecifics in the CA1 subfield of mice. As we discuss in the Discussion section of our manuscript, we feel that this is not necessarily in contradiction with previous reports of socialPCs in rats and bats, since in those experiments the observer was mostly passive, making the

distinction between socialPCs, alloSVCs and egoSVCs rather difficult. In addition, as pointed by reviewer 1 (point 3), potential differences across species or gender could also contribute to harmonize both sets of results.

5. Again in Supplementary Figure 4, no statistically significant distinction in the cross-validated decoding error between social-place cells and the shuffled data is observed.

The difference between Figure 1 and what is now Figure S5 (previously Figure S4) is that in the latter case data are grouped by mouse. Again, we observe that while the improvement is very small, in almost all mice the decoded error is smaller than the corresponding shuffled error (Wilcoxon matched-pairs signed rank test p : 0.0015). Our interpretation is that although this group of cells is very small, if enough of them are used a decoder could have an accurate idea of the absolute position of the conspecific relative to the environment.

6. Kindly explain the criteria used to pinpoint the specific moments for calculating allocentric and egocentric social-vector cells based on head orientation. I am particularly interested in understanding the chosen temporal instances during which the mouse is actively encoding spatial information related to conspecifics, while excluding intervals associated with alternate behaviors such as foraging and sniffing.

We agree with the reviewer that dissecting the specific moments in which social-vector coding is relevant to CA1 would be of great interest. However, in this first characterization our intention was to make a broad and unbiased assessment, so we included the activity of cells across the whole session. This included instances of interaction but also instances of more self-centered behavior such as grooming or foraging. The fact that social-vector representations were found even without filtering for instances of interaction speaks in favor of the robustness of the finding.

Our results comparing the representation of familiar vs unfamiliar conspecifics (Fig. 3) partially address the question posed by the reviewer. We observed that mice spent around 3-fold longer times interacting with unfamiliar mice. However, the proportion of cells classified as egoSVC or alloSVC were similar, populations overlapped significantly and differences in information and decoding error were small. These observations suggest that the overall effect of interaction times on egoSVC and alloSVC maps is weak, perhaps because physical interactions occur only at very small distances, which represents a rather small portion of the effective arena. Note that we found a similar proportion of fields at all covered distances, pointing to the possibility that these cells monitor the position of others regardless of whether or not there is active or imminent interaction. We feel that the fact that most interaction is associated with small distances between mice makes a deeper dissection of the effects of interaction tricky without introducing a completely new behavioral paradigm, something that we are interested in developing in the future.

7. In line 169, the authors mentioned that “coding of conspecifics is not directly related to the amount of interaction within the environment”. To enhance the comprehension of this statement, it would be beneficial to provide a more detailed explanation, as the manuscript does not currently offer conclusive evidence.

This comment refers to ‘directly’ in opposition as ‘inversely’ but we now see that it is a rather obscure phrasing. We have rephrased the text to make this point clearer:

“... suggesting that coding of conspecifics could be influenced by prior experience and does not necessarily improve with the amount of interaction within the environment.”

Minor comments:

1. The manuscript lacks specific information regarding the GRIN lenses employed in the study. By checking Supplementary Figure 1, it is likely that there are variations in lens diameters, with some appearing to be approximately 1mm in diameter, while others appear to be closer to 2mm. I kindly request clarification on this aspect.

We agree that this was not very clear in our original manuscript. All lenses are 2 mm in diameter and apparent differences in the size of the trace observable in histological sections are due to the fact that lenses were not always equally centered on the area of interest, as is now explained in the caption of Figures S1:

“Note that all GRIN lenses are 2 mm in diameter and that the different size in lens trace is due to the fact that not all lenses were perfectly centered in the region of interest.”

2. Could you please provide clarification regarding the choice of miniscopes used for imaging: UCLA Miniscope V3 or V4?

We have now specified in the methods section that we used UCLA Miniscope V3.

3. In some cases, spread depression can indeed affect calcium imaging activity in the CA1 region when using GRIN lenses and Miniscope systems. I am interested in whether this potential issue was encountered or addressed during the course of this study.

We agree with the reviewer. We only observed signs of spread depression in two animals and discarded them, as is now made explicit in the Methods section:

“In two animals we found signs of spread depression, in the form of a slowly evolving calcium wave. These animals and their data were excluded from this work.”

4. To enhance clarity, please provide clarification on whether the head orientation of the mice was determined or extracted using the DeepLabCut software in this study.

We used DeepLabCut to estimate head direction from two LEDs attached to the miniscope, and rotating together with the head of the mouse, as is now made explicit in the Methods section:

“DeepLabCut was also used to extract head direction from the location of two LEDs attached to the miniscope, that rotated with the head of the mouse.”

REVIEWER COMMENTS

Reviewer #1 (Remarks to the Author):

The authors have done a significant amount of work in effectively addressing all my primary concerns raised during the initial review. I recommend its acceptance for publication.

Reviewer #2 (Remarks to the Author):

We appreciate the authors' efforts in addressing our concerns. The revised manuscript has certainly improved - the additional analyses, experiments, and clarifications more strongly support their claims.

There is one important point we raised, however, that we do not feel was properly addressed. In our Major Comment #2, we worried that poor sampling of the different reference frames could lead to high false positive rates in the classification of functional cell types. Our specific concern was:

"The sampling is especially problematic because of the potential interaction between the two reference frames. In other words, an animal may visit all places in the environment and all vectors relative to the conspecific but will rarely visit all combinations of the two (e.g., north of the conspecific while being in the south end of the arena) ...This could be addressed by plotting the coverage of the space in several reference frames against each other."

The authors agreed that coverage is important and that it was not sufficiently addressed in the original submission. Their solution was to calculate the percentage of spatial/angular bins that were occupied in each reference frame and to correlate these measures to each other in a pairwise fashion. This does not solve the problem of poor sampling due to the interaction of reference frames. As we wrote in the initial review, a mouse could cover 100% of reference frame 1 and 100% of reference frame 2, yet not evenly sample all bins of one reference frame in the other reference frame. To provide a more concrete toy example, consider the following example of 2D spatial position and head direction coverage: A mouse explores the entire x-y space of the arena by running in clockwise concentric circles with decreasing radius. In other words, they run clockwise around the outside of the arena, then make a slightly tighter circle, and tighter still, etc. until they reach the center. They covered 100% of x bins, 100% of y bins, and 100% of head direction angles. Critically, however, they were never heading west while in the upper part of the arena, and never facing east while in the lower part of the arena. This toy example is extreme, but

these types of interactions are almost surely present in real data. The question is how they impact the false positive rate.

We recommend analyzing the data in the following manner:

1. Create a 3D scatter plot to show the joint coverage for each animal/session: plot the animal's x position on the x-axis, y position on the y-axis, and egoSVC angle on the z-axis. Repeat for x, y, and alloSVC. Calculate the occupancy for each 3D space where the animal visited that 3D bin for a minimum time of, for example, 2 seconds. Make a figure summarizing the results of these occupancy maps in 3D. Report the percentage of the 3D environment covered for each session/variable triplet. Include this and representative examples of both good and bad coverage in a supplemental figure.
2. Simulate cells with firing rate statistics matching the experimental data, but without any egoSVC or alloSVC tuning. This could be done by sampling from a Poisson process based on the spatial firing rate maps of neurons in the dataset. Generate 1000 of such "non-tuned" cells for each behavioral session separately. Classify the cells as egoSVC or alloSVC in the same way as was done for the experimental data. Calculate the percentage of "non-tuned" cells per session that pass this criterion, yielding a false positive rate given the animal's behavior. Plot the coverage of 3D space against the false positive rate for egoSVC tuning (repeat for alloSVC). Attempt to artificially increase the occupancy by concatenating a few sessions to show how false positive rates fluctuate with 60 min of data compared with 50, 40, 30, 20, or 10 min.

If the false positive rates remain within an acceptable range (perhaps < 10%) given the typical amount of data recorded per session, our concerns would be largely alleviated. If a few sessions exceed this rate, they could be excluded. If many sessions significantly exceed this rate, the conclusions need to be appropriately adjusted. We feel strongly that these analyses must be done before publication.

Reviewer #3 (Remarks to the Author):

The authors have successfully addressed the majority of the previously raised concerns and have enriched their manuscript with additional figures to substantiate their conclusions. The manuscript now exhibits improved organization and clarity in its rewritten form. However, I would recommend addressing the following queries to ensure comprehensive evaluation and robustness of the findings:

Q1: In the spatial maps presented, such as in Figure 1e (inclusive of selfPC, socialPC, alloSVC, and egoSVC), is there a uniform MAX value utilized for normalization across all cells, or is normalization conducted on a cell-by-cell basis?

Q2: (Line 320, Page 15): The manuscript posits that the hippocampal CA1 subfield can represent a single external body from multiple perspectives. Would it enhance the accuracy of your findings to decode the position of a single external body using an amalgamation of various cell types?

Q3: (Line 356, Page 17): The study mentions improvements in the tuning of individual social-vector cells. Given the advent of advanced imaging techniques, such as a wider field of view miniscope, how might the ability to image more cells (including selfPC, socialPC, alloSVC, and egoSVC) impact the results? Would this lead to more precise decoding abilities, or would it primarily expand the place field coverage in an open field arena?

Q5: Please include detailed product information regarding the GRIN lenses used in your experiments.

Q6: (Figure 1J and Figure 2g): The decoding performance of SocialPC appears indistinct from the Shuffle in Figure 1J and similarly in Figure 2g. Could you provide an explanation or interpretation for this observation?

Q7: (Figure 6d): In Figure 6d, the baseline correlation for Naive conditions is noted to be around 0.55.

REVIEWER COMMENTS

Reviewer #1 (Remarks to the Author):

The authors have done a significant amount of work in effectively addressing all my primary concerns raised during the initial review. I recommend its acceptance for publication.

Reviewer #2 (Remarks to the Author):

We appreciate the authors' efforts in addressing our concerns. The revised manuscript has certainly improved - the additional analyses, experiments, and clarifications more strongly support their claims.

There is one important point we raised, however, that we do not feel was properly addressed. In our Major Comment #2, we worried that poor sampling of the different reference frames could lead to high false positive rates in the classification of functional cell types. Our specific concern was:

"The sampling is especially problematic because of the potential interaction between the two reference frames. In other words, an animal may visit all places in the environment and all vectors relative to the conspecific but will rarely visit all combinations of the two (e.g., north of the conspecific while being in the south end of the arena) ...This could be addressed by plotting the coverage of the space in several reference frames against each other."

The authors agreed that coverage is important and that it was not sufficiently addressed in the original submission. Their solution was to calculate the percentage of spatial/angular bins that were occupied in each reference frame and to correlate these measures to each other in a pairwise fashion. This does not solve the problem of poor sampling due to the interaction of reference frames. As we wrote in the initial review, a mouse could cover 100% of reference frame 1 and 100% of reference frame 2, yet not evenly sample all bins of one reference frame in the other reference frame. To provide a more concrete toy example, consider the following example of 2D spatial position and head direction coverage: A mouse explores the entire x-y space of the arena by running in clockwise concentric circles with decreasing radius. In other words, they run clockwise around the outside of the arena, then make a slightly tighter circle, and tighter still, etc. until they reach the center. They covered 100% of x bins, 100% of y bins, and 100% of head direction angles. Critically, however, they were never heading west while in the upper part of the arena, and never facing east while in the lower part of the arena. This toy example is extreme, but these types of interactions are almost surely present in real data. The question is how they impact the false positive rate.

We recommend analyzing the data in the following manner:

1. Create a 3D scatter plot to show the joint coverage for each animal/session: plot the animal's x position on the x-axis, y position on the y-axis, and egoSVC angle on the z-axis. Repeat for x, y, and alloSVC. Calculate the occupancy for each 3D space where the animal visited that 3D bin for a minimum time of, for example, 2 seconds. Make a figure summarizing the results of these occupancy maps in 3D. Report the percentage of the 3D environment covered for each session/variable triplet. Include this and representative examples of both good and bad coverage in a supplemental figure.
2. Simulate cells with firing rate statistics matching the experimental data, but without any egoSVC or alloSVC tuning. This could be done by sampling from a Poisson process based on the spatial firing rate maps of neurons in the dataset. Generate 1000 of such "non-tuned" cells for each behavioral session separately. Classify the cells as egoSVC or alloSVC in the same way as was done for the experimental data. Calculate the percentage of "non-tuned" cells per session that pass this criterion, yielding a false positive rate given the animal's behavior. Plot the coverage of 3D space against the false positive rate for egoSVC tuning (repeat for alloSVC). Attempt to artificially increase the occupancy by concatenating a few sessions to show how false positive rates fluctuate with 60 min of data compared with 50, 40, 30, 20, or 10 min.

If the false positive rates remain within an acceptable range (perhaps < 10%) given the typical amount of data recorded per session, our concerns would be largely alleviated. If a few sessions exceed this rate, they could be excluded. If many sessions significantly exceed this rate, the conclusions need to be appropriately adjusted. We feel strongly that these analyses must be done before publication.

The reviewer raises some crucial concerns that, after careful analysis, have proved valid in some of our data. In summary (we discuss the details below) egoSVC cells, which are the main focus of this manuscript, pass all tests. However, the same tests unveiled that a fraction of alloSVCs are false positives, possibly related to how selfPC place fields close to the border project into the alloSVC effective arena in a biased way. We have added a new supplementary Figure explaining this overestimation of alloSVCs. We have made modifications highlighting the predominant role played by egoSVCs, which has only reinforced the focus that previous versions of our manuscript already had on egoSVCs (among main Figures, only Figure 1 showed otherPCs and alloSVCs).

1. Supplementary Figure 5a now shows 3D scatter plots of self-position vs alloSVC and egoSVC angle, together with quantifications of coverage (cutoff: 1 s). To contextualize these quantifications of coverage, we plotted the percentage of bins with an occupation of at least 1 s as a function of bin size, or more specifically as a function of the total number of bins filling the 3D volume for different bin sizes. This representation allowed for a direct comparison of the 3D coverage with coverage in the selfPC space, where we also plotted covered bins vs number of bins filling the 2D area for different bin sizes. All curves showed a decrease in coverage with decreasing bin size. However, while selfPC and egoSVC curves were practically overlapping, coverage in alloSVC space was always lower, which indicates systematic poor coverage and relates to results below.

2. The proposed analysis is ingenious and eventually led us to interesting discoveries. However, when we applied it exactly as proposed we got a 2-fold increase in the average fraction of 'reconstructed' selfPCs (62.5% vs the original value of 32.3%) and alloSVCs (19.1% vs the original value of 10.7%). We believe that there are two main issues with this reconstruction. First, unlike spikes in electrophysiology, the calcium signal (or its deconvolution) does not follow any known statistics (see for example the methods of Shuman et al, Nature Neuroscience, 2020). Second, correlations between first and second halves of the session, used as a classification criterion, increased in reconstructed cells due to the loss of the original temporal structure. To tackle these issues, we opted for a shuffling-in-place procedure, which we believe maintains the spirit of the reviewer's suggestion. We divided the arena in 10 cm bins and shuffled data inside each bin, otherwise maintaining the dynamics of the calcium signal intact. The size of spatial bins used for shuffling-in-place (not for maps) is a compromise between maximizing variability across shuffle instances and invariance of selfPC representations. As shown in Supplementary Fig. 5c, the average percentage of shuffled cells classified as selfPCs was similar to the original one (33.7% vs 32.3%). The percentage of egoSVCs decreased from 20.0% to 3.3%, but the percentage of alloSVCs only changed from an original value of 10.7% to 10.2%. We also compared the original information with the shuffled one: 92% of egoSVCs had information rates higher than the 95th percentile of the shuffle-in-place distribution (1000 shuffles), but only 42% of alloSVCs, also supporting the idea that a fraction of alloSVCs are false positives.

In Supplementary Figure 5e we show some examples of alloSVCs in the selfPC and alloSVC coordinates, which explain the false positive mechanism. Roughly, when a selfPC has a field close to the left border of the physical arena, this activity selectively projects to the right side of the effective alloSVC arena (because the other mouse cannot stand at the left of the left border), which can sometimes result in a field with enough information for a classification as alloSVC. In the same panel we also show examples of alloSVC maps that cannot be explained by a selfPC field. More generally, Supplementary Fig. 5e-f shows that out of the cells originally classified as alloSVC, those with a selfPC center of mass close to the border of the arena tend to produce shuffled false positives (classified as alloSVC in more than half of shuffles-in-place), while those with a selfPC center of mass close to the center of the arena tend not to. Supplementary Fig. 5g-h shows that if only alloSVCs that do not produce shuffle-in-place false positives are taken into account (137 out of 2162 cells, or 6%), decoding error is lower than shuffle, which also applies to egoSVCs. This observation suggests that although alloSVCs exhibit many false positives, they still represent a legitimate population, albeit smaller than we originally thought. Supplementary Fig. 5d also shows that the percentage of false positives in alloSVCs does not correlate with coverage. This observation is reasonable if one thinks that these false positives are not produced by anecdotal poor coverage, but by a systematic lack of coverage along the borders of the physical arena that relates to the previous point and to the lower curve for alloSVC in Supplementary Fig. 5b.

In our manuscript, alloSVCs are characterized in supplementary Figures. In each one of them, we have added key analyses including only the subset of alloSVCs that do not produce a majority of shuffle-in-place false positives, mostly showing similar trends.

In the results section we have introduced the following:

In contrast, the high overlap between selfPC and alloSVC, together with geometrical considerations, suggests that a fraction of cells classified as alloSVC are false positives (Supplementary Fig. 5).

We have also added in the first paragraph of the discussion:

“Our main result is that the representation of others in CA1 is multiplexed, but mostly done through egocentric social-vector coding”

and in the second one:

“, although not all representations are equally predominant,”

“The first group, which proved hard to assess from free exploration experiments given a bias toward false positive classification from place cells firing close to the border of the environment,”

“, and the widely dominant one in terms of number of dedicated cells,”

, while the last sentence speculating on the reasons of the overlap of alloSVCs with other cell types was removed.

We have also modified the abstract to highlight the role of egoSVCs and tone down that of alloSVCs. As mentioned before, previous versions of our manuscript already focused on egoSVCs and these changes only reinforced this focus.

Reviewer #3 (Remarks to the Author):

The authors have successfully addressed the majority of the previously raised concerns and have enriched their manuscript with additional figures to substantiate their conclusions. The manuscript now exhibits improved organization and clarity in its rewritten form. However, I would recommend addressing the following queries to ensure comprehensive evaluation and robustness of the findings:

Q1: In the spatial maps presented, such as in Figure 1e (inclusive of selfPC, socialPC, alloSVC, and egoSVC), is there a uniform MAX value utilized for normalization across all cells, or is normalization conducted on a cell-by-cell basis?

As the reviewer points out, the color scale goes from 0 to MAX, where MAX is different across neurons. The value that MAX takes for each neuron, or the peak event rate in the map, is indicated as P above each map. To clarify this point we have modified the legend in the colorbar, which now says “Max. (P)” and also added a short phrase in the captions:

“Peak event rate (P; red in maps) and spatial information (I) ...”

Q2: (Line 320, Page 15): The manuscript posits that the hippocampal CA1 subfield can represent a single external body from multiple perspectives. Would it enhance the accuracy of your findings to decode the position of a single external body using an amalgamation of various cell types?

This is an intriguing question. We show in this manuscript that the hippocampus uses in parallel different coordinates systems to represent a conspecific. While decoding within each independent reference frame does not benefit from information in other reference frames, whether or not these different information pathways synergistically converge downstream is a key question that we cannot address with the current data. A valid alternative is that all these representations are simultaneously available to the mouse so it can use whichever is most useful for the task that it is facing. Social interactions make use of space in a range of ways. The observation that egoSVCs do not change in number but achieve sharper tuning through training in a task that requires them supports this idea of simultaneous availability, although future work more precisely intended to dissect between these alternatives is required before jumping into conclusions.

Q3: (Line 356, Page 17): The study mentions improvements in the tuning of individual social-vector cells. Given the advent of advanced imaging techniques, such as a wider field of view miniscope, how might the ability to image more cells (including selfPC, socialPC, alloSVC, and egoSVC) impact the results? Would this lead to more precise decoding abilities, or would it primarily expand the place field coverage in an open field arena?

As the reviewer indicates, recent developments such as that of the MiniLFOV (Guo et al, Science Advances, 2023), will potentially allow to record more cells simultaneously from a single animal. We believe that in an arena of the same size social vector decoding could improve substantially, and perhaps achieve levels similar to those of place decoding, if more cells were simultaneously available. At some point, however, adding more cells might not lead to substantial improvements in decoding, as discussed by Hazon and colleagues (2022, Nature Communications). In the case of social vector cells, Figure S6c supports the idea that decoding can still improve, and in this sense they behave very similarly to place cells. In contrast, the same figure seems to suggest that socialPC decoding does not benefit from having more neurons. As throughout our manuscript, socialPC coding, until now the only proposed candidate mechanism for representing others in the hippocampus, has a very poor decoding performance, although significantly higher than chance.

Q5: Please include detailed product information regarding the GRIN lenses used in your experiments.

We have now specified in the Methods the use of Edmund Optics 1.8 mm diameter and 4.31 mm long GRIN lenses (670 nm, 0 mm working distance). We believe these lenses are manufactured by GrinTech, but this is not specified by the vendor.

Q6: (Figure 1J and Figure 2g): The decoding performance of SocialPC appears indistinct from the Shuffle in Figure 1J and similarly in Figure 2g. Could you provide an explanation or interpretation for this observation?

In Figure 1j socialPCs seem to perform at chance levels but actually the statistical test shows that they do slightly better than chance. The effect size, however, is very small. This is very important because so far socialPCs were the only candidate mechanism proposed to code for others in the hippocampus, and we are showing that this form of coding is very poor in mice behaving freely, and in a pursuit task. In contrast, social vector cells seem to be much more promising as a mechanism to mediate social behavior.

Figure 2g deals with the decoding of egoSVC position of others relative to chance levels. The decoding is significantly better than chance both when training a decoder with egoSVC cells for the same or for a different mouse, while using the same mouse is slightly better than using a different mouse. We understand that the reviewer is surprised by the fact that a decoder trained with the same mouse performs so similarly to a decoder trained with a different mouse. We believe that this happens because there is some shared code or generalization across mice, as suggested by Figures 2d,f. Although the number of neurons in the shared code seems to be rather low, these neurons could perhaps be better encoders. However, it is important to note that decoding error falls exponentially and not linearly with decoding resources, as shown in Supplementary Figure 5c, so perhaps the difference between decoding with egoSVCs related to the same or the other mouse is not so small as it visually looks like.

Q7: (Figure 6d): In Figure 6d, the baseline correlation for Naive conditions is noted to be around 0.55.

This rather high value of correlation reflects the fact that neurons are not only active in their main egoSVC field but are also very active throughout the environment. As a consequence, the correlation of the population code between two opposite ends of the maze is quite high and has a lot of room for improvement. This improvement certainly takes place through learning of the task. EgoSVCs in trained animals have a much larger fraction of their calcium events inside their main egoSVC field (Fig. 6c), and as a consequence the baseline correlation falls to around 0.2.

REVIEWERS' COMMENTS

Reviewer #2 (Remarks to the Author):

The authors have successfully addressed our concerns and we appreciate the effort. It's great to see that the additional analyses helped identify potential false positives in the cell classification and clarify the predominant role of egocentric tuning.

We have some suggestions that we believe would add clarity, but ultimately leave it to the authors' discretion.

1. In the response to reviewers, the authors wrote "the percentage of alloSVCs only changed from an original value of 10.7% to 10.2%," suggesting that the true percentage of alloSVCs is 0.5%. But below they wrote "if only alloSVCs that do not produce shuffle-in-place false positives are taken into account (137 out of 2162 cells, or 6 %)." Where is the 6% coming from? Regardless, with an alpha level of 0.05, we strongly disagree with the interpretation that alloSVCs represent a legitimate cell population. This is especially problematic because the main conclusion of the paper, emphasized in prominent places like the title and abstract, is that the representation of others is multiplexed. If the percentages of alloSVCs and socialPCs are negligible, then there is no multiplexing, the conspecifics are simply encoded via an egocentric vector. The authors themselves note that all of the main figures focus on egocentric tuning. The main message and claims should be adjusted to match the results.

2. The percentages of each cell type reported in the main text and figures should have the false positives removed first. There is no reason to report numbers that are known to be inflated by false positives.

3. Supplemental Figure 5c should include bars for actual and shuffled data side-by-side so that it is clear what is being shown. It's not currently obvious that the results shown are for shuffled data from the figure alone. From first glance it actually appears that after correcting for false positives only the alloSVC population survives.

4. In the legend for Supplemental Figure 5e we recommend removing the line "Similar considerations do not apply to egoSVCs." Similar considerations would apply if the behavior of the mice were biased.

REVIEWERS' COMMENTS

Reviewer #2 (Remarks to the Author):

The authors have successfully addressed our concerns and we appreciate the effort. It's great to see that the additional analyses helped identify potential false positives in the cell classification and clarify the predominant role of egocentric tuning.

We have some suggestions that we believe would add clarity, but ultimately leave it to the authors' discretion.

1. In the response to reviewers, the authors wrote "the percentage of alloSVCs only changed from an original value of 10.7% to 10.2%," suggesting that the true percentage of alloSVCs is 0.5%. But below they wrote "if only alloSVCs that do not produce shuffle-in-place false positives are taken into account (137 out of 2162 cells, or 6 %)." Where is the 6% coming from? Regardless, with an alpha level of 0.05, we strongly disagree with the interpretation that alloSVCs represent a legitimate cell population. This is especially problematic because the main conclusion of the paper, emphasized in prominent places like the title and abstract, is that the representation of others is multiplexed. If the percentages of alloSVCs and socialPCs are negligible, then there is no multiplexing, the conspecifics are simply encoded via an egocentric vector. The authors themselves note that all of the main figures focus on egocentric tuning. The main message and claims should be adjusted to match the results.

We agree that this point was not completely clear in our previous response. From an original classification of 10.7% of alloSVCs, there is a 6% of cells that is not classified as alloSVC in more than half of the shuffles-in-place. These cells cannot be discarded as false positives, and some examples are shown in Supplementary Figure 6e. Among the 10.2% of cells classified as alloSVC through shuffle-in-place, there are many cells that were not originally classified as alloSVC, so one cannot simply subtract the two percentages.

Regarding the idea of multiplexed coding vs purely egocentric coding, we prefer to be cautious. Although we prove in this manuscript that egocentric social-vector coding is the main way in which CA1 represents others, there are still significant populations of alloSVCs and even socialPCs. A direct binomial comparison with a random process with probability of 0.05 would be correct if our classification criterion was based only on information levels surpassing the 95th percentile of the shuffled distribution. However, we also include the criterion of stability between halves of the session, by means of which many informative cells are prevented from being classified positively. As an independent argument, decoding is better than chance levels for socialPCs and for alloSVCs (with or without false positives), which would not be expected from a random distribution. For these reasons, and given that the reviewer leaves it to our discretion, we think that the best description of our results is of a multiplexed code with prevalence of egoSVC representations.

We have clarified these points in Supplementary Figure 6:

“While the initial classification suggested that alloSVCs constitute on average 10.7% of all cells, incorporating the supplementary criterion of more than half of shuffles-in-place not being classified as alloSVC reduces the proportion to 6% of cells.”

2. The percentages of each cell type reported in the main text and figures should have the false positives removed first. There is no reason to report numbers that are known to be inflated by false positives.

We have included a reference to the false-positive figure together with the reporting of the percentage:

“The category with the highest percentage of cells was selfPC (42.7 %), followed by egoSVC (13.1 %) and alloSVC (11.1 %, but see Supplementary Fig. 6) (Fig. 1g).”

3. Supplemental Figure 5c should include bars for actual and shuffled data side-by-side so that it is clear what is being shown. It's not currently obvious that the results shown are for shuffled data from the figure alone. From first glance it actually appears that after correcting for false positives only the alloSVC population survives.

We agree and have included both panels side by side, in what is now Supplementary Figure 6c (previously Supplementary Figure 5c).

4. In the legend for Supplemental Figure 5e we recommend removing the line “Similar considerations do not apply to egoSVCs.” Similar considerations would apply if the behavior of the mice were biased.

We agree and have removed the phrase in what is now Supplementary Figure 6e (previously Supplementary Figure 5e).

Reviewer #3 (Remarks to the Author):

The authors have diligently addressed all the concerns outlined during the review. Based on their thorough revisions and improvements, I confidently recommend the acceptance of this paper for publication.